# ALIGNING PERSISTENT HOMOLOGY
# WITH GRAPH POOLING

## ABSTRACT

Recently, there has been an emerging trend to integrate persistent homology (PH) into graph neural networks (GNNs) to enrich expressive power. However, naively plugging PH features into GNN layers always results in marginal improvement with low interpretability. In this paper, we investigate a novel mechanism for injecting global topological invariance into pooling layers using PH, motivated by the observation that filtration operation in PH naturally aligns graph pooling in a cut-off manner. In this fashion, message passing in the coarsened graph is performed along persistent sub-topology, leading to improved performance. Experimentally, we apply our mechanism to a collection of graph pooling methods and observe consistent and substantial performance gain over several popular datasets, demonstrating its wide applicability and flexibility. Code is open-sourced at `https://anonymous.4open.science/r/TIP`.

## 1 INTRODUCTION

Persistent homology (PH) is a powerful tool in the field of topological data analysis, which is capable of evaluating stable topological invariant properties from unstructured data in a multi-resolution fashion (Edelsbrunner & Harer, 2022). Concretely, PH derives an increasing sequence of simplicial complex subsets by applying a filtration function (see Fig. 1(a)). According to the fact that PH is at least as expressive as Weisfeiler-Lehman (WL) hierarchy (Horn et al., 2021), there recently emerged a series of works seeking to merge PH into graph neural networks (GNNs), delivering competitive performance on specific tasks (Wong & Vong, 2021; Zhao et al., 2020; Horn et al., 2021). Standard schemes of existing works achieve this by employing pre-calculated topological features (Zhao et al., 2020) or placing learnable filtration functions in the neural architectures (Hofer et al., 2020; Horn et al., 2021). Such integration of PH features is claimed to enable GNNs to emphasize persistent topological sub-structures. However, it is still unclear to what extent the feature-level integration of PH is appropriate and how to empower GNNs with PH other than utilizing features.

Graph pooling (GP) in parallel plays an important role in a series of graph learning methods (Grattarola et al., 2022), which hierarchically aggregates an upper-level graph into a more compact lower-level graph. Typically, GP relies on calculating an assignment matrix taking into account local structural properties such as community (Müller, 2023) and cuts (Bianchi et al., 2020). Though the pooling paradigm in convolutional neural networks (CNNs) is quite successful (Krizhevsky et al., 2012), some researchers raise concerns about its effectiveness and applicability in graphs. For example, Mesquita et al. (2020) challenges the local-preserving usage of GP by demonstrating that random pooling even leads to similar performance. Till now, it remains opaque what property should be preserved for pooled topology to better facilitate the downstream tasks.

From Fig. 1(a), it is readily observed that PH and GP both seek to coarsen/sparsify a given graph in a hierarchical fashion: while PH gradually derives persistent sub-topology (substructures that have meaningful topology) by adjusting the filtering parameter, GP obtains a sub-graph by performing a more aggressive cut-off. In a sense of understanding a graph through a hierarchical lens, PH and GP turn out to align with each other well.

Driven by this observation, in this paper, we investigate the mechanism of aligning PH and GP so as to mutually reinforce each other. To this end, we conduct experiments by running a pioneer GP method DiffPool (Ying et al., 2018) to conduct graph classification on several datasets and at the same time use the technique in Hofer et al. (2020) to compute PH information. We manually

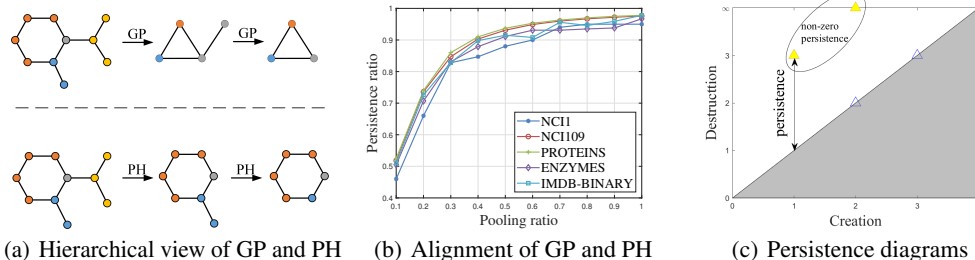

(a) Hierarchical view of GP and PH    (b) Alignment of GP and PH    (c) Persistence diagrams

Figure 1: Illustration of Graph Pooling (GP) and Persistent Homology (PH). (a) GP and PH share a similar hierarchical fashion by coarsening a graph. (b) As a motivating experiment, we gradually change pooling ratio and count how persistence ratio (ratio of non-zero persistence) changes with it. (c) Illustration of persistence diagrams.

change pooling ratio and see what proportion of meaningful topological information (characterized by the ratio of non-zero persistence) is naturally preserved at the final training stage. Surprisingly, the correspondence is quite stable regardless of different datasets (see Fig. 1(b)), which implies the monotone trend between the pooling ratio and non-zero persistence is commonly shared by a large range of graph data. As a consequence, we develop a natural way to integrate PH and GP in both feature and topology levels. Concretely, in addition to concatenating vectorized PH diagram as supplementary features, we further enforce the coarsened graph to preserve topological information as much as possible with a specially designed PH-inspired loss function. Hence we term our method Topology-Invariant Pooling (TIP). TIP can be flexibly injected into a variety of existing GP methods, and demonstrates a consistent ability to provide substantial improvement over them. We summarize our contributions as follows:

- We for the first time investigate the way of aligning PH with GP, by investigating the monotone relationship in between.

- We further design an effective mechanism to inject PH information into GP at both feature and topology levels, with a novel topology-preserving loss function.

- Our mechanism can be flexibly integrated with a variety of GP methods, achieving consistent and substantial improvement over multiple datasets.

## 2 RELATED WORK

**Graph pooling.** Graph pooling has been used in various applications, which can reduce the graph size while preserving its structural information. Early methods are based on clustering to coarsen graphs, such as the greedy clustering method Graclus (Dhillon et al., 2007), non-negative matrix factorization of the adjacency matrix (Bacciu & Di Sotto, 2019), and spectral clustering (Ma et al., 2019). Recently, learnable graph pooling methods have gained popularity, which learn to select important nodes in an end-to-end manner. DiffPool (Ying et al., 2018) follows a hierarchical learning structure by utilizing GNNs to learn clusters and gradually aggregate nodes into a coarser graph. MinCutPool (Bianchi et al., 2020) optimizes a normalized cut objective to partition graphs into clusters. DMoNPool (Müller, 2023) optimizes the modularity of graphs to ensure high-quality clusters. SEP (Wu et al., 2022) generates clusters in different hierarchies simultaneously without compromising local structures. These methods are classified as dense pooling due to the space complexity they incur. Despite their effectiveness, dense pooling methods have been criticized for high memory cost and complexity (Cangea et al., 2018). Therefore, various sparse pooling methods have been proposed, such as Top-K (Gao & Ji, 2019), ASAPool (Ranjan et al., 2020), and SAGPool (Lee et al., 2019). These methods coarsen graphs by selecting a subset of nodes based on a ranking score. As they drop some nodes in the pooling process, these methods are criticized for their limited capacity to retain essential information, with potential effects on the expressiveness of preceding GNN layers (Bianchi & Lachi, 2023).

**Persistent homology in GNNs.** PH is a technique to calculate topological features of structured data, and many approaches have been proposed to use PH in graph machine learning due to the high expressiveness of topological features on graphs (Hofer et al., 2017). Since isomorphic graphs may exhibit different topological features, the combination of PH and the Weisfeiler-Lehman (WL) algorithm leads to stronger expressive power (Rieck et al., 2019). This encourages further exploration on equipping GNNs with topological features. Zhao et al. (2020) propose that message passing in GNNs can be effectively reweighted using topological features. Hofer et al. (2020) and Horn et al. (2021) provide theoretical and practical insights that filtraions in PH can be purely learnable, enabling flexible usage of topological features in GNNs. However, existing methods tend to view PH merely as a tool for providing supplementary information to GNNs, resulting in only marginal improvements and limited interpretability.

## 3 BACKGROUND

We briefly review the background of this topic in this section, as well as elaborate on the notations.

Let $\mathcal{G} = (V, E)$ be an undirected graph with $n$ nodes and $m$ edges, where $V$ and $E$ are the node and the edge sets, respectively. Nodes in attributed graphs are associated with features, and we denote by $V = \{(v, \mathbf{x}_v)\}_{v \in 1:n}$ the set of nodes $v$ with $d$ dimensional attribute $\mathbf{x}_v$. It is also practical to represent the graph with an adjacency matrix $\mathbf{A} \in \{0, 1\}^{n \times n}$ and the node feature matrix $\mathbf{X} \in \mathbb{R}^{n \times d}$

**Graph Neural Networks.** We focus on the general message-passing GNN framework that updates node representations by iteratively aggregating information from neighbors (Gilmer et al., 2017). Concretely, the $k$-th layer of such GNNs can be expressed as:

$$\mathbf{X}^{(k)} = \mathrm{M}\left(\mathbf{A}, \mathbf{X}^{(k-1)}; \theta^{(k)}\right) \tag{1}$$

where $\theta^{(k)}$ is the trainable parameter, and $\mathrm{M}$ is the message propagation function. Numbers of $\mathrm{M}$ have been proposed in previous research (Kipf & Welling, 2016; Hamilton et al., 2017). A complete GNN is typically instantiated by stacking multiple layers of Eq. 1. Hereafter we denote by $\mathrm{GNN}(\cdot)$ an arbitrary such multi-layer GNN for brevity.

**Dense Graph Pooling.** GP in GNNs is a special layer designated to produce a coarsened or sparsified sub-graph. Formally, GP can be formulated as $\mathcal{G} \mapsto \mathcal{G}_P = (V_P, E_P)$ such that the number of nodes $|V_P| \leq n$. GP layers can be placed into GNNs in a hierarchical fashion to persistently coarsen the graph. Typical GP approaches (Ying et al., 2018; Bianchi et al., 2020; Müller, 2023) rely on learning a soft cluster assignment matrix $\mathbf{S}^{(l)} \in \mathbb{R}^{n_{l-1} \times n_l}$:

$$\mathbf{S}^{(l)} = \mathrm{softmax}\left(\mathrm{GNN}^{(l)}\left(\mathbf{A}^{(l-1)}, \mathbf{X}^{(l-1)}\right)\right). \tag{2}$$

Subsequently, the coarsened adjacency matrix at the $l$-th pooling layer is calculated as

$$\mathbf{A}^{(l)} = \mathbf{S}^{(l)\top}\mathbf{A}^{(l-1)}\mathbf{S}^{(l)} \tag{3}$$

and the corresponding node representations are calculated as

$$\mathbf{X}^{(l)} = \mathbf{S}^{(l)\top}\mathrm{GNN}^{(l)}\left(\mathbf{A}^{(l-1)}, \mathbf{X}^{(l-1)}\right). \tag{4}$$

These approaches differ from each other in the way to produce $\mathbf{S}$, which is used to inject a bias in the formation of clusters. In our work, we select three GP methods, i.e., DiffPool (Ying et al., 2018), MinCutPool (Bianchi et al., 2020), and DMoNPool (Müller, 2023), to cope with. Details of these methods are in Appendix A.

**Topological Features of Graphs.** Simplicial complexes stand as the focal point within the realm of algebraic topology. An assembly of simplices of certain dimensions constitutes a simplicial complex denoted as $K$. A graph can be seen as a low-dimensional simplicial complex that only contains 0-simplices (vertices) and 1-simplices (edges) (Horn et al., 2021). The simplest kind of topological features describing graphs are Betti numbers, formally denoted as $\beta_0$ for the number of connected components and $\beta_1$ for the number of cycles.

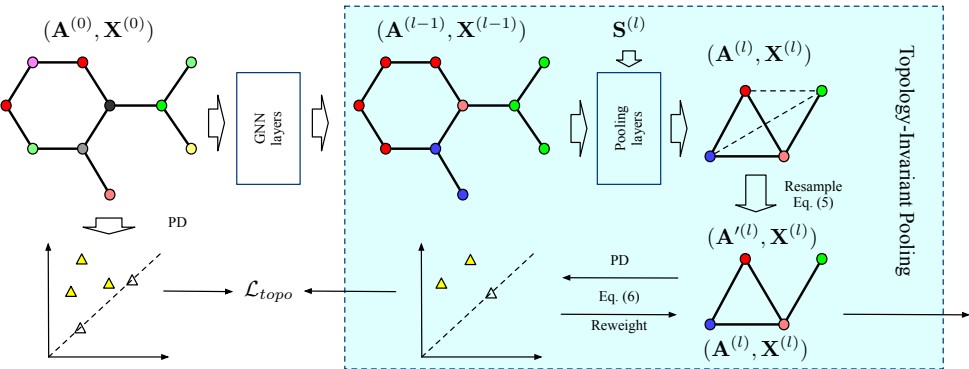

Figure 2: Overview of our method, where the shaded part corresponds to one layer of Topology-Invariant Pooling.

Despite the limited expressive power of these two numbers, it can be improved by evaluating them alongside a filtration. Filtrations are scalar-valued functions of the form $f : V \cup E \to \mathbb{R}$ , which assigns each vertice and edge a value. Changes in the Betti numbers, named as persistent Betti numbers, can subsequently be monitored throughout the progress of the filtration: by considering a threshold ($a \in \mathbb{R}$), we can analyze the subgraph originating from the pre-image of $((-\infty, a])$ of $f$, denoted as $(f^{-1}((-\infty, a]))$. The image of $f$ leads to a set of node values $a_1 < \cdots < a_n$ and generates a sequence of nested subgraphs of the form $\emptyset \subseteq \mathcal{G}_0 \subseteq \ldots \mathcal{G}_k \ldots \subseteq \mathcal{G}_n = \mathcal{G}$, where $\mathcal{G}_k = (V_k, E_k)$ is a subgraph of $\mathcal{G}$ with $V_k := \{v \in V \mid f(\mathbf{x}_v) \leq a_k\}$ and $E_k := \{(v, w) \in E \mid \max\{f(x_v), f(x_w)\} \leq a_k\}$. This process is also known as persistent homology (denoted as $ph(\cdot)$) on graphs. Typically, persistent Betti numbers are characterized in a persistence diagram (PD) as $\mathrm{ph}(\mathcal{G}, f) = \{\mathcal{D}_0, \mathcal{D}_1, ...\}$, which is made up of tuples $(a_i, a_j) \in \mathbb{R}^2$, with $a_i$ and $a_j$ representing the creation and destruction of a topological feature respectively (see Fig. 1(c)). The absolute difference in function values $|a_j - a_i|$ is called the persistence of a topological feature, where high persistence corresponds to features of the function, while low persistence is typically considered as noise (Horn et al., 2021; Rieck, 2023). Moreover, we follow the settings in previous works (Horn et al., 2021; Hofer et al., 2017) to extend $\mathcal{D}_1$ as follows: (1) each cycle is paired with the edge that created it; (2) edges $e$ that do not create a cycle (still in this circle) are assigned a 'dummy' tuple value, such as $(f(e), f(e))$; (3) all other edges will be paired with the maximum value of the filtration. Therefore, $\mathcal{D}_1$ consists of as many tuples as the number of edges $m$.

## 4 METHODOLOGY

### 4.1 OVERVIEW

An overview of our method is shown in Fig. 2, where the shaded part corresponds to one layer of Topology-Invariant Pooling. The upper part is the GP process and the lower part is the injection of PH. Let $(\mathbf{A}^{(0)}, \mathbf{X}^{(0)})$ be the input graph. We consider to perform a GP at the $(l-1)$-th layer. After obtaining a coarsened (densely connected) graph $(\mathbf{A}^{(l)}, \mathbf{X}^l)$ with a standard GP method, we resample the coarsened graph using Gumbel-softmax trick as $\mathbf{A}'^{(l)}$ in order to make it adapt to PH. Then, this coarsened graph is further reweighted injecting persistence, and is optimized by minimizing the topological gap $\mathcal{L}_{topo}$ from the original graph, yielding $(\mathbf{A}^{(l)}, \mathbf{X}^l)$. By stacking multiple TIP layers, hierarchical pooling emphasizing topological information can be achieved. In the following sections, we elaborate on the detailed design of our mechanism.

### 4.2 TOPOLOGY-INVARIANT POOLING

In many real-world applications, such as molecular graphs analysis (Swenson et al., 2020; Hofer et al., 2020), topological features of graphs are of utmost importance. However, typical GNNs fail to capture certain topological structures in graphs, such as cycles (Bouritsas et al., 2022; You et al., 2021; Huang et al., 2022). Moreover, in dense graph pooling, graphs are pooled without preserving any topology. Even if we manage to make GNN topology-aware, the coarsened graph has no meaningful

topology at all, impairing the use of GNNs in these tasks. To overcome these limitations, we propose to inject topological information into GP. We resort to PH to characterize the importance of edges. Note that for those edges do not form cycles, their creation and destruction are the same, leading to zero persistence.

The core of PH is notion of filtration, which presents a challenging task to choose the right filtration. As the coarsened graph evolves in each training step, integrating PH into GP demands multiple computations of filtrations. To address this, we place learnable filtration (LF) functions for incorporating PH information, which is flexible and efficient as demonstrated by Hofer et al. (2020). LF relies on node features and graph topology, which are readily available in GP. Consequently, LF can be seamlessly integrated into GP with minimal computational overhead. Specifically, we employ an MLP network $\Phi(\cdot)$ as the filtration function together with $\mathrm{sigmoid}(\cdot)$ to map node features $\mathbf{X} \in \mathbb{R}^{n \times d}$ into $n$ scalar values. Recently, an increasing amount of attention has been devoted to cycles (Bouritsas et al., 2022; You et al., 2021; Huang et al., 2022) due to their significant relevance to downstream tasks in various domains such as biology (Koyutürk et al., 2004), chemistry (Murray & Rees, 2009), and social network analysis (Jiang et al., 2010). Recognizing that cycles offer an intuitive representation of graph structure, we focus on the one-dimensional PD associated with cycles. Following the standard way in GP (Eq. 2 3 4), we additionally propose the subsequent modules to inject PH into GP at both feature and topology levels.

**Resampling.** One major limitation of utilizing LF is that the computation process is unconscious of edge weights, i.e. edges with non-negative weights will be treated equally, so PH cannot directly extract meaningful topological features from $\mathbf{A}^{(l)}$. Besides, rethinking GP in Eq. 3, the coarsened adjacency matrix has limited expressive power for two reasons. First, although $\mathbf{S}^{(l)}$ is a soft assignment matrix obtained by $\mathrm{softmax}(\cdot)$, each element still has nonzero values, i.e. $\mathbf{A}^{(l)}$ is always densely connected. Second, the edge weights may span a wide range by multiplication (see Appendix D for empirical evidence). These drawbacks hinder the stability and generalization power of the subsequent message passing layers (Gong & Cheng, 2019). None of existing GP methods can handle these problems properly.

Therefore, we resample the coarsened adjacency $\mathbf{A}^{(l)}$ obtained from a normal GP layer (Eq. 3) as:

$$\mathbf{A}'^{(l)} = \mathrm{resample} \left( \frac{\mathbf{A}^{(l)} - \min(\mathbf{A}^{(l)})}{\max(\mathbf{A}^{(l)}) - \min(\mathbf{A}^{(l)})} \right) \tag{5}$$

where $\mathbf{A}^{(l)}$ is first normalized in the range of $[0, 1]$, and $\mathrm{resample}(\cdot)$ is performed independently for each matrix entry using the Gumbel-softmax trick (Jang et al., 2016). In practice, only the upper triangular matrix is resampled to make it symmetric and we add self-loops to the graph.

**Persistence Injection.** Now $\mathbf{A}'^{(l)} \in \{0, 1\}^{n_l \times n_l}$ is a sparse matrix without edge features so we can easily inject topological information into it. For a resampled graph with $\mathbf{A}'^{(l)}$ and $\mathbf{X}^{(l)}$, we formulate the persistence injection as:

$$\mathcal{D}_1 = \mathrm{ph}(\mathbf{A}'^{(l)}, \mathrm{sigmoid}(\Phi(\mathbf{X}^{(l)}))), \qquad \mathbf{A}^{(l)} = \mathbf{A}'^{(l)} \odot \mathrm{to\_dense}(\mathcal{D}_1[1] - \mathcal{D}_1[0]) \tag{6}$$

where $\odot$ is the Hadamard product, $\mathrm{to\_dense}()$ means transforming sparse representations in terms of edges to dense matrix representations, $\mathcal{D}_1[i]$ is the $i$-th value in each tuple of $\mathcal{D}_1$, and we denote the updated adjacency matrix after persistence injection still as $\mathbf{A}^{(l)}$ for notation consistency. Persistence injection can actually be regarded as a reweighting process. Since the filtration values are within $[0, 1]$, $\mathbf{A}^{(l)}$ after persistence injection is guaranteed to have edge weights in the range of $[0, 1]$ and is passed to the next pooling layer.

**Topological Loss Function.** The aforementioned mechanism can explicitly inject topological information into graphs, but it relies on the condition that the coarsened graph retains certain essential sub-topology. To this end, we propose an additional loss function to guide the GP process. Intuitively, the coarsened graph should exhibit similarity to the original graph in terms of topology. Since the computation of PH is differentiable, one possible approach is to directly minimize the differences between the PDs of the original graph and the coarsened graph. However, this implementation would require computing the Wasserstein distance between two PDs through optimal transport (Yan et al., 2022), which is intractable in training due to its complexity. Considering that our objective is to

estimate the difference, we instead propose vectorizing the PDs and minimizing their high-order statistical features (Okabe et al., 2018). Specifically, we use several transformations (denoted as $\text{transform}(\cdot)$) and concatenate the output, including the triangle point transformation, the Gaussian point transformation and the line point transformation introduced in Carrière et al. (2020) to convert the tuples in PD into vector $\mathbf{h}_t$ ($t \in [1, m]$). We calculate the mean vector $\mu$ as well as the second-order statistics as the standard deviation vector $\sigma$ as:

$$\mathbf{h}_t = \text{transform}(\mathcal{D}_1), \qquad \mu = \frac{1}{m} \sum_{t=1}^{m} \mathbf{h}_t, \qquad \sigma = \sqrt{\frac{1}{m} \sum_{t=1}^{m} \mathbf{h}_t \odot \mathbf{h}_t - \mu \odot \mu} \qquad (7)$$

In this manner, the difference between two PDs can be estimated through the comparison of their statistics in the features, which is the concatenation of the mean and variance vectors. To further regularize the topological difference between layers, we introduce a topological loss term defined as:

$$\mathcal{L}_{\text{topo}} = \frac{1}{Ld} \sum_{l=1}^{L} \sum_{i=1}^{d} \left( \left( \mu_i^{(l)} \| \sigma_i^{(l)} \right) - \left( \mu_i^{(0)} \| \sigma_i^{(0)} \right) \right)^2 \qquad (8)$$

where $(\cdot \| \cdot)$ stands for the concatenation operation, $L$ is the number of pooling layers, and $d$ is the feature dimension. Note that the intuition behind $\mathcal{L}_{topo}$ is different from the loss functions in existing graph pooling methods: the coarsened graph after pooling should be topologically similarly to the original graph rather than having exact cluster structures.

### 4.3 ANALYSIS

In this section we examine the validity of our proposed method, and in particular analyze its expressive power and complexity.

**Theorem 1** *The 1-dimensional topological features computed by persistence homology is sufficient enough to be at least as expressive as 1-WL in terms of distinguishing isomorphic graphs with self-loops, i.e. if the 1-WL label sequences for two graphs $\mathcal{G}$ and $\mathcal{G}'$ diverge, there exists an injective filtration $f$ such that the corresponding 1-dimensional persistence diagrams $\mathcal{D}_1$ and $\mathcal{D}_1'$ are not equal.*

This result demonstrates that the 1-dimensional topological features contain sufficient information to potentially perform at least as well as 1-WL when it comes to distinguishing non-isomorphic graphs. We can then obtain the concluding remark that TIP is more expressive than other dense pooling methods by showing that there are pairs of graphs that cannot be distinguished by 1-WL but that can be distinguished by TIP.

**Proposition 1.** *TIP is isomorphic invariant.*

Detailed proof and illustrations of the theorem and proposition can be found in Appendix C.

**Complexity.** Persistent homology can be efficiently computed for dimensions 0 and 1, with a worst-case time complexity of $O(m\alpha(m))$, where $m$ represents the number of sorted edges in a graph. Here, $\alpha(\cdot)$ represents the inverse Ackermann function, which is extremely slow-growing and can essentially be considered as a constant for practical purposes. Therefore, the primary factor that affects the calculation of PH is the complexity of sorting all the edges, which is $O(m \log m)$. Our resampling and persistence encoding mechanism ensures that the coarsened graphs are sparse rather than dense, making our approach both efficient and scalable.

## 5 EXPERIMENTS

In the experiments, we evaluate the benefits of persistent homology on several state-of-the-art graph pooling methods, with the goal of answering the following questions:

**Q1.** Is PH capable of preserving topological information during pooling?

**Q2.** How does PH affect graph pooling in preserving task-specific information?

To this end, we showcase the empirical performance of TIP on two tasks, namely, topological similarity (Section 5.2) and graph classification (5.3). Our primary focus is to assess in which scenarios topology can enhance GP. We further conduct ablation study to investigate the contributions of different modules, which are shown in Appendix E.2.

## 5.1 EXPERIMENTAL SETUP

**Models.** To investigate the effectiveness of PH in GP, we integrate TIP with DiffPool, MinCutPool, and DMoNPool, which are the pioneering approaches that have inspired many other pooling methods. Additionally, as most pooling methods rely on GNNs as their backbone, we compare the widely used GNN models GCN(Kipf & Welling, 2016), GIN(Xu et al., 2018), and GraphSage (Hamilton et al., 2017). We also look into another two related and State-of-the-Art GNN models, namely TOGL (Horn et al., 2021) and GSN (Bouritsas et al., 2022), which incorporate topological information and graph substructures into GNNs to enhance the expressive power. Furthermore, we compare several other GP methods, namely Graclus (Dhillon et al., 2007) and TopK (Gao & Ji, 2019). For model selection, we follow the guidelines provided by the original authors or benchmarking papers. Our method acts as an additional plug-in to existing pooling methods (referred to as -TIP) without modifying the remaining model structure and hyperparameters. Appendix B.1 provides detailed configurations of these models.

**Datasets.** To evaluate the capabilities of our model across diverse domains, we assess its performance on a variety of graph datasets commonly used in graph classification tasks. We select seven benchmarks from TU datasets (Morris et al., 2020) and one benchmark from OGB datasets (Hu et al., 2020). Specifically, we adopt molecular datasets NCI1, NCI109, and OGBG-MOLHIV, bioinformatics datasets ENZYMES, PROTEINS, and DD, as well as social network datasets IMDB-BINARY and IMDB-MULTI. We use the default dataset settings (i.e. the number and type of features) from PyG library [1]. Furthermore, to investigate the topology-preserving ability of our method, we conduct experiments on several highly structured datasets (ring, torus, grid2d) obtained from the PyGSP library [2]. Appendix B.2 provides detailed statistics of the datasets.

## 5.2 PRESERVING TOPOLOGICAL STRUCTURE

In this experiment, we study Q1 about the ability of PH to preserve topological structure during pooling. Specifically, we assess the topological similarity between the original and coarsened graphs $\mathcal{G}$ and $\mathcal{G}'$, by comparing the Wasserstein distance associated with their respective PDs $\mathcal{D}_1$ and $\mathcal{D}'_1$. This evaluation criterion is widely used to compare graphs of different sizes (Wong & Vong, 2021; Yan et al., 2022). To calculate PD, we utilize Forman curvature on each edge of the graph as the filtration, which incorporates edge weights and graph clusters to better capture the topological features of the coarsened graphs (Sreejith et al., 2016; Wee & Xia, 2021). Specifically, we consider the 1-Wasserstein distance $W\left(\mathcal{D}_1, \mathcal{D}'_1\right) = \inf_{\gamma \in \Pi\left(\mathcal{D}_1, \mathcal{D}'_1\right)} \mathbb{E}_{(x,y)\sim\gamma}[\|x-y\|]$ as the evaluation metric, where $\Pi(\cdot)$ is the set of joint distributions $\gamma(x, y)$ whose marginals are $\mathcal{D}_1$ and $\mathcal{D}'_1$, respectively. Note that we are not learning a new filtration but keep a fixed one. Rather, we use learnable filtrations in the training process to enhance flexibility, and solely optimize $L_{topo}$ as the objective.

We compare TIP with other pooling methods. Table 1 reports the average $W$ values on three datasets, demonstrating that TIP can improve dense pooling methods to a large margin and have the best topological similarity. We visualize the pooling results in Fig. 3 for better interpretation, where isolated nodes with no links are omitted for clarity. It is evident that DiffPool, MinCutPool, and DMoNPool tend to generate dense graphs and fail to preserve any topological structures. Conversely, our method, which incorporates topological features using PH, sparsifies the coarsened graphs and reveals certain essential topological structures. Notably, in the ring and torus datasets, large cycles are clearly preserved by our method. It is also interesting to observe that the grid2d dataset, despite having a different spatial layout, exhibits similar topology to torus (with four adjacent nodes forming a small cycle), resulting in similar shapes of their corresponding coarsened graphs. This indicates that the objective function indeed contributes to preserving topological similarity to some extent. Sparse

---

[1]`https://pytorch-geometric.readthedocs.io/en/latest/generated/torch_geometric.datasets.TUDataset.html`

[2]`https://pygsp.readthedocs.io/en/stable/reference/graphs.html`

Table 1: Results to show the topology-preserving ability of different pooling methods. We show the Wasserstein distance (the smaller, the better) to assess the topological similarity. A **bold** value indicates the overall winner.

| Methods | Datasets | | |
|---|---|---|---|
| | ring | torus | grid2d |
| Graclus | $37.62 \pm 4.41$ | $124.47 \pm 12.07$ | $35.82 \pm 0.93$ |
| TopK | $14.24 \pm 1.06$ | $35.15 \pm 4.78$ | $84.12 \pm 2.21$ |
| DiffPool | $234.57 \pm 9.49$ | $237.89 \pm 20.66$ | $146.91 \pm 6.05$ |
| DiffPool-TIP | $\mathbf{8.03 \pm 3.08}$ | $17.97 \pm 2.19$ | $\mathbf{32.26 \pm 3.21}$ |
| MinCutPool | $232.60 \pm 10.81$ | $248.51 \pm 15.69$ | $155.16 \pm 21.79$ |
| MinCutPool-TIP | $18.11 \pm 5.59$ | $\mathbf{11.38 \pm 2.21}$ | $58.71 \pm 9.84$ |
| DMoNPool | $224.48 \pm 22.25$ | $236.97 \pm 16.54$ | $142.85 \pm 27.53$ |
| DMoNPool-TIP | $16.10 \pm 4.80$ | $17.34 \pm 4.76$ | $52.26 \pm 5.75$ |

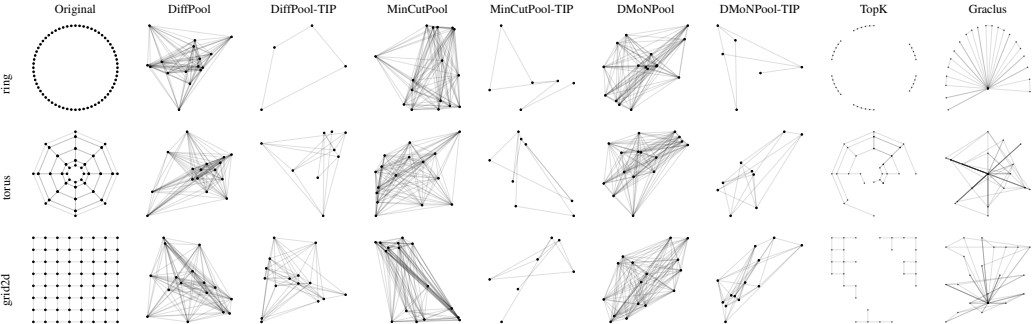

Figure 3: Coarsened graphs from different methods in the preserving topological structure experiment.

pooling methods, which tend to preserve local topology, perform slightly better than the original dense pooling methods.

## 5.3 GRAPH CLASSIFICATION

In this experiment, we examine the impact of PH on GP in downstream tasks to answer **Q2**. We have observed in the former experiment that PH can preserve essential topological information during pool. However, two additional concerns arise: (1) Does TIP continue to generate invariant sub-topology in the downstream task? (2) If so, does this sub-topology contribute to the performance of the downstream task? To address these concerns, we evaluate TIP using various graph classification benchmarks, where the accuracy achieved on these benchmarks serves as a measure of a method's ability to selectively preserve crucial information based on the task at hand.

We begin by visualizing the coarsened graphs in this task, where edges are cut-off by a small value. From Fig. 4, we can clearly observe that our method manage to preserve the essential sub-topology similar to the original graphs, while dense pooling methods cannot preserve any topology. As discussed in Mesquita et al. (2020), dense pooling methods, such as DiffPool, achieve comparable performance when the assignment matrix $\mathbf{S}$ is replaced by a random matrix. Here our visualization reveals that regardless of the value of $\mathbf{S}$, the coarsened graph always approaches to a fully connected one. Sparse pooling methods, on the other hand, manage to preserve some local structures through clustering or dropping, but the essential global topological structures are destroyed.

Table 2 presents the average and standard deviation of the graph classification accuracy on benchmark datasets. Additionally, the results of several baseline GNNs are provided. Experimental results demonstrate that TIP can consistently enhance the performance of the three dense pooling methods. While the original dense pooling methods sometimes underperform compared to the baselines, they are able to surpass them after integrating TIP. Among the different variants of dense pooling methods,

Table 2: Test accuracy of graph classification on benchmark datasets. A **bold** value indicates the overall winner. Gray background indicates that TIP outperforms the base GP.

| Methods | Datasets | | | | | | | |
| --- | --- | --- | --- | --- | --- | --- | --- | --- |
| | NCI1 | NCI109 | ENZYMES | PROTEINS | DD | IMDB-BINARY | IMDB-MULTI | OGBG-MOLHIV |
| GCN | 77.81 ± 1.50 | 74.90 ± 1.85 | 32.51 ± 3.35 | 76.65 ± 3.14 | 78.66 ± 2.36 | 74.20 ± 2.40 | 53.23 ± 3.04 | 75.04 ± 0.84 |
| GIN | 80.30 ± 1.70 | 79.66 ± 1.55 | 42.83 ± 3.66 | 77.18 ± 3.35 | 78.05 ± 3.60 | 72.65 ± 3.04 | 53.28 ± 3.16 | 76.03 ± 0.84 |
| GraphSage | 80.85 ± 1.25 | 79.16 ± 1.28 | 39.17 ± 3.28 | 76.67 ± 3.05 | 78.83 ± 3.07 | 76.60 ± 2.37 | 53.46 ± 2.39 | 76.18 ± 1.27 |
| TOGL | 80.53 ± 2.29 | 78.27 ± 1.39 | 46.09 ± 3.72 | 78.17 ± 2.80 | 76.10 ± 2.24 | 76.65 ± 2.75 | 53.87 ± 2.67 | 77.21 ± 1.33 |
| GSN | **83.50 ± 2.00** | N / A | N / A | 74.59 ± 5.00 | N / A | **76.80 ± 2.00** | 52.60 ± 3.60 | 76.06 ± 1.74 |
| Graclus | 80.82 ± 1.27 | 79.13 ± 1.79 | 41.44 ± 3.46 | 75.69 ± 2.62 | 74.67 ± 2.45 | 74.45 ± 3.29 | 54.72 ± 2.79 | 76.81 ± 0.70 |
| TopK | 79.43 ± 3.50 | 77.96 ± 1.58 | 38.35 ± 4.83 | 76.03 ± 2.94 | 76.97 ± 3.94 | 72.60 ± 4.24 | 53.66 ± 2.93 | 76.28 ± 0.67 |
| DiffPool | 77.64 ± 1.86 | 76.50 ± 2.32 | 48.34 ± 5.14 | 78.81 ± 3.12 | 80.27 ± 2.51 | 73.15 ± 3.30 | 54.32 ± 2.99 | 76.60 ± 1.04 |
| DiffPool-TIP | 83.02 ± 1.70 | **81.09 ± 1.65** | **65.05 ± 4.24** | **79.86 ± 3.12** | **82.12 ± 2.53** | 76.40 ± 3.13 | **55.53 ± 2.92** | **77.75 ± 1.18** |
| MinCutPool | 77.92 ± 1.67 | 75.88 ± 2.06 | 39.83 ± 2.63 | 78.25 ± 3.84 | 79.15 ± 3.51 | 73.80 ± 3.54 | 53.87 ± 2.95 | 75.60 ± 0.54 |
| MinCutPool-TIP | 80.17 ± 1.29 | 79.48 ± 1.37 | 46.34 ± 3.85 | 79.73 ± 3.27 | 80.87 ± 2.47 | 75.20 ± 2.67 | 54.47 ± 2.27 | 77.18 ± 0.83 |
| DMoNPool | 78.03 ± 1.64 | 76.62 ± 1.94 | 40.82 ±3.68 | 78.63 ± 3.89 | 79.16 ± 3.61 | 73.50 ± 3.01 | 54.07 ± 3.08 | 76.30 ± 1.34 |
| DMoNPool-TIP | 79.68 ± 1.38 | 78.46 ± 1.50 | 45.84 ± 5.32 | 79.73 ± 3.66 | 81.46 ± 2.96 | 74.25 ± 3.06 | 54.23 ± 2.64 | 76.70 ± 0.62 |

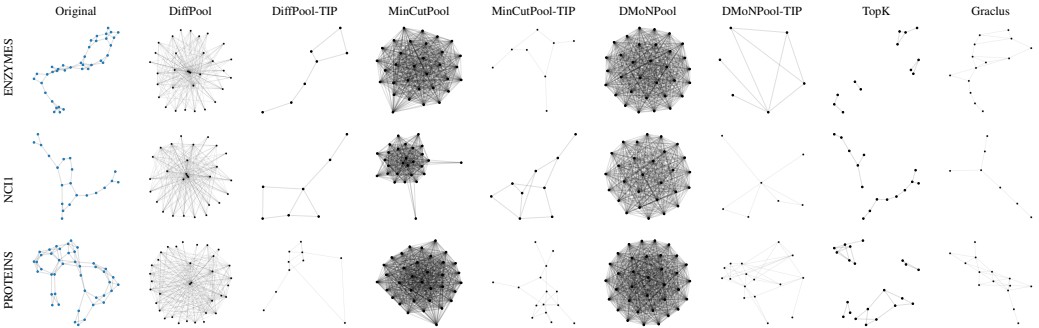

Figure 4: Graphs pooled with different methods in graph classification experiment.

DiffPool-TIP achieves the highest accuracy in most cases, this may be attributed to the fact that DiffPool applies three consecutive layers of GNNs after each pooling operation, while the other two methods only utilize one GNN layer. The coarsened graphs with invariant sub-topology are mostly sparsely connected, so additional layers of GNN may have a positive impact on passing crucial messages.

Moreover, an intriguing observation can be found on ENZYMES, where TOGL significantly surpasses the baseline GNNs. TOGL in practice, incorporates PH into GNNs, so this results underscores the significance of incorporating topological information for improved performance on ENZYMES. Similarly, our method also demonstrates notable improvements by augmenting the three dense pooling methods on the ENZYMES dataset. However, it is worth noting that TOGL only exhibits marginal improvements or even underperforms on the other datasets. This suggests that simply integrating PH features into GNN layers does not fully exploit topological information. Conversely, injecting global topological invariance into pooling layers yields superior performance. Lastly, we provide the training curves in Appendix E.4, which demonstrate that incorporating meaningful topology leads to improved performance.

## 6 CONCLUSION

In this paper, we developed a method named Topology-Invariant Pooling (TIP) that effectively integrates global topological invariance into graph pooling layers. This approach is inspired by the observation that the filtration operation in PH naturally aligns with the GP process. We theoretically showed that PH is at least as expressive as WL-test, with evident examples demonstrating TIP's expressivity beyond dense pooling methods. Empirically, TIP indeed preserved persistent global topology information, and achieved substantial performance improvement on top of several pooling methods on various datasets, demonstrating strong flexibility and applicability.

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

Table 3: Unsupervised loss functions of graph pooling

| Method | $\mathcal{L}_r$ | $\mathcal{L}_c$ |
|---|---|---|
| DiffPool | $\left\|\mathbf{A}, \mathbf{SS}^T\right\|_F$ | $\frac{1}{n}\sum_{i=1}^{n} H\left(\mathbf{S}_i\right)$ |
| MinCutPool | $-\frac{\mathrm{Tr}\left(\mathbf{S}^\top \mathbf{AS}\right)}{\mathrm{Tr}\left(\mathbf{S}^\top \mathbf{DS}\right)}$ | $\left\|\frac{\mathbf{S}^\top \mathbf{S}}{\|\mathbf{S}^\top \mathbf{S}\|_F} - \frac{\mathbf{I}_C}{\sqrt{C}}\right\|_F$ |
| DMoNPool | $-\frac{1}{2m} \cdot \mathrm{Tr}\left(\mathbf{S}^\top \mathbf{BS}\right)$ | $\left\|\frac{\mathbf{S}^\top \mathbf{S}}{\|\mathbf{S}^\top \mathbf{S}\|_F} - \frac{\mathbf{I}_C}{\sqrt{C}}\right\|_F + \frac{\sqrt{C}}{n}\left\|\sum_i \mathbf{C}_\mathbf{i}^\top\right\|_F - 1$ |

## A  DENSE GRAPH POOLING METHODS

Generally, dense graph pooling methods follow a hierarchical architecture, but their motivations differ. DiffPool suggests that nearby nodes should be pooled together, drawing on insights from link prediction and the assignment matrix $\mathbf{S}$ should be approximate to a one-hot vector so that the clusters are less overlapped with each other. MinCutPool, on the other hand, adapts the normalized cut as a regularizer for pooling. This encourages strongly connected nodes to be pooled together, ensures orthogonal cluster assignments, and promotes clusters of similar size. Moreover, DMoNPool additionally proposes a regularization to optimize the modularity quality of clusters so that the pooling can generate high quality clusters approach to ground truth. In summary, each of these methods introduces two types of unsupervised loss functions: the reconstruction loss $\mathcal{L}_r$, which regulates how the coarsened graph is reconstructed to retain some cluster structure, and the other is the cluster loss $\mathcal{L}_c$, which prevents convergence to local minima. The detailed formulations of these loss functions are provided in Table 3, where $||\cdot||_F$ denotes the Frobenius norm, $H$ denotes the entropy function, $\mathbf{S}_i$ is the $i$-th row of $\mathbf{S}$, $\mathbf{D}$ is the degree matrix, $C$ is the number of clusters, $\mathbf{B} = \mathbf{A} - \frac{\mathbf{DD}^T}{2m}$ is the modularity matrix, respectively.

## B  EXPERIMENTAL SETUP

### B.1  IMPLEMENTATION DETAIL

**Evaluation.**  In the graph classification task, all datasets are splitted into train (80%), validation (10%), and test (10%) data. Following the evaluation protocol in Ying et al. (2018); Mesquita et al. (2020), we train all models using the Adam optimizer (Kingma & Ba, 2014) and implement a learning rate decay mechanism, reducing the learning rate from $10^{-3}$ to $10^{-5}$ with a decay ratio of 0.5 and a patience of 10 epochs. Additionally, we use early stopping based on the validation accuracy with patience of 50 epochs. We report statistics of the performance metrics over 20 runs with different seeds.

**Hyperparameters.**  For dense pooling methods, the pooling ratio ranges from $[0.1, 0.5]$, the number of pooling layers is 2, and the hidden dimension is selected from $\{32, 64\}$. For the Graclus method we use 2 pooling layers, while for TopK we use 3 pooling layers with a pooling ratio of 0.8. The batch size for all models is uniformly set to 20, and the maximum number of training epochs is 1000. For the graphs obtained from the PyGSP library (ring, torus, grid2d), the number of nodes in each graph is fixed at 64.

**Model configuration.**  All the methods are implemented using PyTorch and PyG (Paszke et al., 2017; Fey & Lenssen, 2019). The compared methods are implemented following the implementations provided in the PyG library [3]. In the case of DiffPool, it uses a 3-layer GraphSage in each pooling layer, while MinCutPool and DMoNPool use a 1-layer GCN before pooling and a 1-layer GNN (Morris et al., 2019) in each pooling layer. Note that in DiffPool, the GNNs in Eqs. 2 and 4 are different, while in MinCut and DNoNPool they are the same one, as what their do in the original papers. TopK and Graclus are based on a 1-layer GNN (Morris et al., 2019). TOGL is implemented using a 3-layer GraphSage as it has demonstrated superior performance on graph classification tasks (see Table 2). For the baseline GNN models (GCN, GIN, and GraphSage), we use 3 layers with mean/max pooling. In our model, TIP is incorporated as a plugin to existing pooling methods, without

---
[3]https://pytorch-geometric.readthedocs.io/en/latest/modules/nn.html

Table 4: Statistics of datasets

| Dataset | #Graphs | #Avg.Nodes | #Avg.Edges | #Features | #Classes |
|---|---|---|---|---|---|
| ENZYMES | 600 | 32.63 | 62.14 | 18 | 6 |
| PROTEINS | 1113 | 39.06 | 72.82 | 3 | 2 |
| NCI1 | 4110 | 29.87 | 32.30 | 37 | 2 |
| NCI109 | 4127 | 29.68 | 32.13 | 38 | 2 |
| DD | 1060 | 232.9 | 583 | 89 | 2 |
| IMDB-BINARY | 1000 | 19.8 | 193.1 | 0 | 2 |
| IMDB-MULTI | 1500 | 13 | 65.94 | 0 | 3 |
| OGBG-MOLHIV | 41127 | 25.5 | 27.5 | 9 | 2 |

modifying the remaining model structure and hyperparameters. We replace the reconstruction loss $\mathcal{L}_r$ with $\mathcal{L}_{topo}$ while keeping the cluster loss $\mathcal{L}_c$ unchanged. In the case of MinCutPool and DMoNPool, our resampling strategy is added after their original normalization of the coarsened graphs. In preserving topological structure experiments, we initialize node features as the concatenation of the first ten eigenvectors of graph Laplacian matrices.

## B.2    DATASET STATISTICS

The statistics of datasets used in this paper are summarized in Table 4, where we show the number of graphs, average number of nodes, average number of edges, number of features, and number of classes.

## C    THEORETICAL EXPRESSIVITY OF TIP

**Theorem 1.** *The 1-dimensional topological features computed by persistence homology is sufficient enough to be at least as expressive as 1-WL in terms of distinguishing isomorphic graphs with self-loops, i.e. if the 1-WL label sequences for two graphs $\mathcal{G}$ and $\mathcal{G}'$ diverge, there exists an injective filtration $f$ such that the corresponding 1-dimensional persistence diagrams $\mathcal{D}_1$ and $\mathcal{D}_1'$ are not equal.*

*Proof.* Assume that $\mathcal{G}$ and $\mathcal{G}'$ have $n$ and $n'$ nodes, and the label sequences of them diverge at some iteration $h$, which means there exists at least one label whose count is unique. Let nodes $u$ and $u'$ be the nodes with unique count in $\mathcal{G}$ and $\mathcal{G}'$, respectively. Denote $La^{(h)} := \{l_1, l_2, ...\}$ as an enumeration of the finitely many hashed labels at iteration $h$. We can build a filtration function $f$ by assigning a vertex $v$ with label $l_i$ to its index, i.e. $f(v) := i$ except that $f(u) = n + n' + 1$ and $f(u') = n + n' + 2$. The filtration of edge $(u, w)$ is defined as $f(v, w) := max\{f(v), f(w)\}$, and for isolated nodes $v$, the filtration of self-loop edges is $f(v, v) = f(v)$. Therefore, node with unique label count and its connected edges always correspond to the largest filtration value. Note that the 1-dimensional PD has been extended to have the same cardinality as the number of edges. If node $u$ or $u'$ forms a circle, the creation of this circle is related to the edge with the largest filtration; if node $u$ or $u'$ does not form a circle, the corresponding edges lie on the diagonal of $\mathcal{D}_1$ with unique coordinates; otherwise node $u$ constitute a circle while $u'$ does not, then the corresponding edges lie in different parts in $\mathcal{D}_1$ and $\mathcal{D}_1'$. Hence, $\mathcal{D}_1 \neq \mathcal{D}_1'$.

To demonstrate that TIP is more expressive than other dense pooling methods, we provide examples of graph pairs that cannot be distinguished by 1-WL but can be by TIP. We present an example of such isomorphic graphs in Fig. 5, where in the second graph the edge connecting two triangles does not form a circle. This edge corresponds to zero persistence and is eliminated in TIP. Consequently, the two originally isomorphic graphs can be easily distinguished. Provided that the three sufficient conditions proposed in (Bianchi & Lachi, 2023) are satisfied, the pooling layers retain the same level of expressive power as GNN. In TIP, the reduction of node features remains unaltered, thereby fulfilling the three conditions. Additionally, TIP is capable of distinguishing certain isomorphic graphs, indicating its superior expressive power compared to conventional dense pooling methods such as DiffPool, MinCutPool, and DMoNPool.

**Proposition 1.** *TIP is isomorphic invariant.*

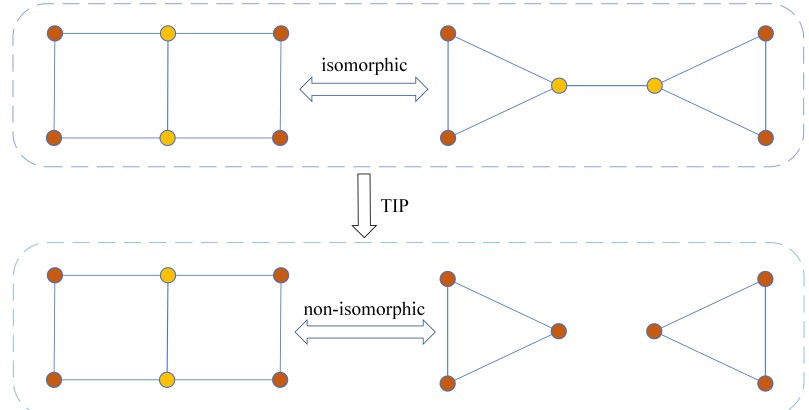

Figure 5: A pair of isomorphic graphs that cannot distinguished by 1-WL but can be distinguished by TIP.

To prove this statement, we adopt the following lemma to show the isomorphic property of PH.

**Lemma 1 (Theorem 2 in Rieck (2023)).** Let $\mathcal{G}_1$ and $\mathcal{G}_2$ be two isomorphic graphs. For any equivariant filtration $f$, the corresponding persistence diagrams are equal.

In TIP, the filtration $f$ is implemented using MLP, ensuring the equivariant property of filtration. Moreover, our resampling operations in Section 4.2 are equivariant. Therefore, the two isomorphic graphs after the resampling and persistence injection operations are still isomorphic to each other. Now we are able to prove Proposition 1.

*Proof.* Let $\mathcal{G}_1$ and $\mathcal{G}_2$ be two isomorphic graphs with $n$ nodes. Their node features after applying several layers of GNNs are denoted as $\mathbf{X}_1 = \{\mathbf{x}_i\}_{i=1}^n$ and $\mathbf{X}_2 = \{\mathbf{y}_i\}_{i=1}^n$, respectively. Let the assignment matrix of $\mathcal{G}_1$ and $\mathcal{G}_2$ obtained in Eq. 2 be $\mathbf{S}_1 = \left\{\{s_{ij}\}_{i=1}^n\right\}_{j=1}^{n'}$ and $\mathbf{S}_2 = \left\{\{q_{ij}\}_{i=1}^n\right\}_{j=1}^{n'}$ with $\sum_{j=1}^{n'} s_{ij} = 1$ and $\sum_{j=1}^{n'} q_{ij} = 1$. Let the coarsened graphs after pooling as $\mathcal{G}_{1P}$ and $\mathcal{G}_{2P}$ and with $n'$ nodes, and their corresponding node features as $\mathbf{X}_{P1} = \{\mathbf{x}_{Pj}\}_{j=1}^{n'}$ and $\mathbf{X}_{P2} = \{\mathbf{y}_{Pj}\}_{j=1}^{n'}$, where $x_{Pj} = \sum_{i=1}^n \mathbf{x}_i \cdot \mathbf{s}_{ij}$ and $y_{Pj} = \sum_{i=1}^n \mathbf{y}_i \cdot \mathbf{q}_{ij}$ according to Eq. 4.

Assume that $\mathcal{G}_1$ and $\mathcal{G}_2$ are isomorphic, then for each pooling layer we have

$$\sum_{i=1}^n \mathbf{x}_i = \sum_{i=1}^n \mathbf{y}_i$$

$$\Rightarrow \sum_{i=1}^n \mathbf{x}_i \cdot \sum_{j=1}^{n'} \mathbf{s}_{ij} = \sum_{i=1}^n \mathbf{y}_i \cdot \sum_{j=1}^{n'} \mathbf{q}_{ij}$$

$$\Rightarrow \sum_{j=1}^{n'} \sum_{i=1}^n \mathbf{x}_i \cdot \mathbf{s}_{ij} = \sum_{j=1}^{n'} \sum_{i=1}^n \mathbf{y}_i \cdot \mathbf{q}_{ij} \tag{9}$$

$$\Rightarrow \sum_{j=1}^{n'} x_{Pj} = \sum_{j=1}^{n'} y_{Pj}$$

which indicates that the output of TIP is invariant given isomorphic graphs as input. We have already proved that isomorphic graph pairs after TIP are still isomorphic, so Eq. 9 still holds for multiple layers of pooling.

## D    EMPIRICAL EVIDENCE

We conduct experiments on the NCI1 dataset and plot the heatmap of the coarsened adjacency matrix in Fig. 6, where we can observe that the edge weights in DiffPool may span a wide range due to the

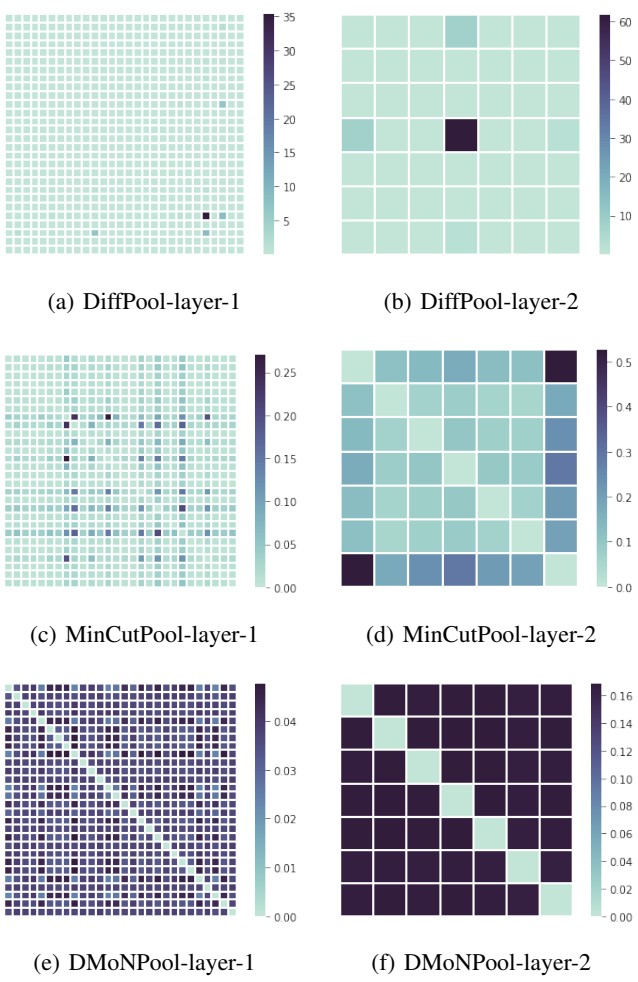

Figure 6: Heatmap of the coarsened adjacency matrix in terms of DiffPool, MinCutPool, and DMoNPool on NCI1 dataset.

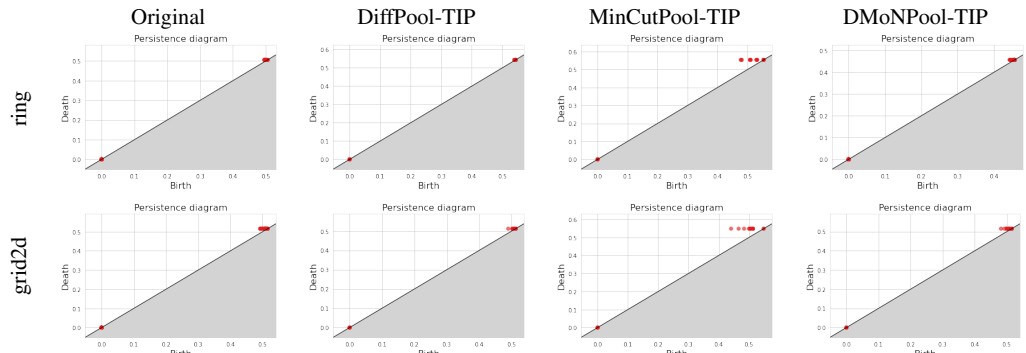

Figure 7: Persistence diagrams of graphs before and after applying TIP in terms of ring and grid2d datasets.

involvement of multiple multiplications in their generation. For MinCutPool and DMoNPool, the edge weights are normalized by degree to mitigate numerical explosion. However, this normalization leads to the edge weights becoming excessively smooth and lacking sparsity. Learnable filtration based PH performs effectively on unweighted graphs; however, none of the existing GP methods are capable of appropriately handling the adjacency matrix.

# E    ADDITIONAL EXPERIMENTS

## E.1    VISUALIZATION OF PERSISTENCE DIAGRAMS

We visually represent the 1-dimensional PD of graphs before and after applying TIP in terms of ring and grid2d datasets, as shown in Fig. 7. As described in Appendix B.1, in the original graphs we initialize node features with the eigenvectors of the graph Laplacian matrices. Consequently, the features of different edges exhibit slight variations, resulting in multiple nonoverlapping points in the PDs. Upon applying TIP, we can clearly observe that the one-dimensional topological features related to cycles remain similar to those in the original graphs. This demonstrates TIP's ability to preserve cycles.

## E.2    ABLATION STUDY

To assess the contributions of different modules in our TIP model, we conducte comprehensive ablation studies on NCI1, PROTEINS, ENZYMES, and IMDB-BINARY datasets. We utilize DiffPool-TIP as the baseline architecture and examined three ablated variants: (i) with no resampling (DiffPool-TIP-NR), (ii) with no persistence injection (DiffPool-TIP-NP), and (iii) with no topological loss function (DiffPool-TIP-NL). As for the ablation of MinCutPool-TIP and DMoNPool-TIP, we adopt a similar naming scheme.

As depicted in Table 5, ablating any of the above modules resulte in performance degradation compared to the full model, thus indicating the importance of each designed module in the success of TIP. Additionally, on all three datasets, the resampling module significantly enhance the classification outcomes, while its removal lead to a substantial performance drop. Without resampling, the learnable filtration will treat edges equally, resulting in the inclusion of nonsensical topological information. In some cases, this even impede the model's performance, as observed in the no injection variants which perform worse than their counterparts on the PROTEINS dataset.

Another noteworthy observation is that even in the absence of the topological loss function $\mathcal{L}_{topo}$, GP can still benefit from incorporating PH. This could be attributed to the fact that the learnable filtration can inherently capture certain essential topological information to some extent. Furthermore, our model can still reap the benefits of the topological loss function, which indirectly guides the pooling process, even without explicitly injecting topological information using persistence.

Table 5: Test accuracy of graph classification in ablation study experiments.

| Methods | Datasets | | | |
|---|---|---|---|---|
| | NCI1 | PROTEINS | ENZYMES | IMDB-BINARY |
| DiffPool | $77.64 \pm 1.86$ | $78.81 \pm 3.12$ | $48.34 \pm 5.14$ | $73.15 \pm 3.30$ |
| DiffPool-TIP-NR | $80.82 \pm 1.71$ | $77.89 \pm 4.07$ | $55.43 \pm 2.81$ | $75.00 \pm 2.64$ |
| DiffPool-TIP-NP | $81.99 \pm 1.15$ | $79.30 \pm 1.26$ | $62.22 \pm 3.13$ | $75.85 \pm 2.85$ |
| DiffPool-TIP-NL | $82.33 \pm 2.14$ | $79.11 \pm 2.01$ | $58.77 \pm 5.15$ | $76.10 \pm 3.78$ |
| DiffPool-TIP | $\mathbf{83.02 \pm 1.70}$ | $\mathbf{79.86 \pm 3.12}$ | $\mathbf{65.05 \pm 4.24}$ | $\mathbf{76.40 \pm 3.13}$ |
| MinCutPool | $77.92 \pm 1.67$ | $78.25 \pm 3.84$ | $39.83 \pm 2.63$ | $73.80 \pm 3.54$ |
| MinCutPool-TIP-NR | $79.68 \pm 1.38$ | $78.23 \pm 2.92$ | $42.51 \pm 2.83$ | $74.35 \pm 1.80$ |
| MinCutPool-TIP-NP | $78.81 \pm 2.07$ | $78.92 \pm 3.35$ | $45.56 \pm 2.81$ | $74.65 \pm 3.24$ |
| MinCutPool-TIP-NL | $78.48 \pm 1.86$ | $78.40 \pm 3.06$ | $45.26 \pm 4.14$ | $74.90 \pm 3.03$ |
| MinCutPool-TIP | $\mathbf{80.17 \pm 1.29}$ | $\mathbf{79.73 \pm 3.27}$ | $\mathbf{46.34 \pm 3.85}$ | $\mathbf{75.20 \pm 2.67}$ |
| DMoNPool | $78.03 \pm 1.64$ | $78.63 \pm 3.89$ | $40.82 \pm 3.68$ | $73.50 \pm 3.01$ |
| DMoNPool-TIP-NR | $79.26 \pm 1.01$ | $78.72 \pm 1.30$ | $42.51 \pm 4.40$ | $73.75 \pm 3.30$ |
| DMoNPool-TIP-NP | $79.60 \pm 0.97$ | $79.44 \pm 1.68$ | $44.36 \pm 3.98$ | $74.50 \pm 3.35$ |
| DMoNPool-TIP-NL | $79.08 \pm 1.83$ | $79.26 \pm 1.70$ | $43.35 \pm 3.90$ | $74.00 \pm 2.76$ |
| DMoNPool-TIP | $\mathbf{79.68 \pm 1.38}$ | $\mathbf{79.73 \pm 3.66}$ | $\mathbf{45.84 \pm 5.32}$ | $\mathbf{74.25 \pm 3.06}$ |

Table 6: Average running time (seconds) comparisons on different datasets.

| Methods | Datasets | | |
|---|---|---|---|
| | NCI1 | PROTEINS | ENZYMES |
| DiffPool | 209.48 | 56.55 | 30.61 |
| DiffPool-TIP | 365.98 | 113.65 | 61.02 |
| MinCutPool | 145.99 | 38.22 | 27.34 |
| MinCutPool-TIP | 341.93 | 102.31 | 55.54 |
| DMoNPool | 124.89 | 35.07 | 19.35 |
| DMoNPool-TIP | 351.42 | 101.86 | 57.91 |

Overall, our ablation study supports the indispensability and effectiveness of each module in the TIP model, further underscoring their contributions to its success.

### E.3 RUNNING TIME COMPARISON

We compare the running time (in seconds) of TIP on different datasets. The experiments are conducted using an AMD EPYC 7542 CPU and a single NVIDIA 3090 GPU. We utilize the default settings from the graph classification experiments. We report the average running time of 50 epoches training in Table 6. It is worth noting that TIP is performed $L$ times for $L$ pooling layers, thus the inclusion of TIP does not impose a significant computational burden.

### E.4 VISUALIZATION OF TRAINING CURVES

In Fig. 8, we present the training objective curve and the wasserstein distance with a fixed filtraion (the same as Section 5.2) on ENZYMES dataset using DiffPool-TIP as an example. The objective value decreases as the coarsened graphs become more similar in topology to the original graphs, indicating that meaningful topology is beneficial for improving performance.

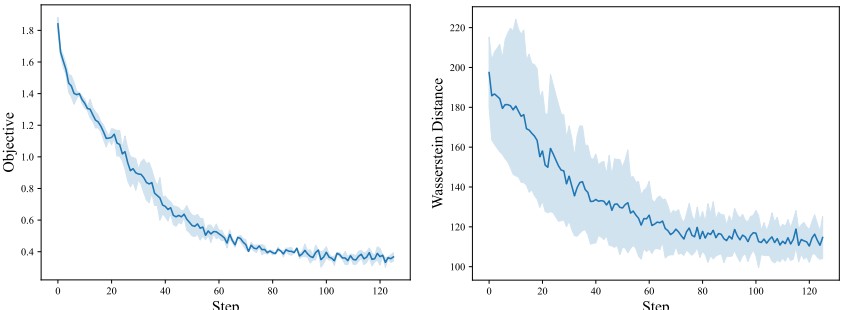

Figure 8: The training curves of DiffPool-TIP on ENZYMES dataset. We show the average values and min-max range in terms of objective and Wasserstein distance for multiple runs.

