# OpenReview forum: "Aligning Persistent Homology with Graph Pooling"
_ICLR.cc/2024/Conference — Submitted to ICLR 2024_

### Official Review · Reviewer_S9nw · 2023-10-28

**Soundness:** 4 excellent
**Presentation:** 3 good
**Contribution:** 3 good
**Rating:** 6
**Confidence:** 3

**Summary:**

In this paper, the authors develop a model called Topology-Invariant Pooling (TIP) to improve the pooling process in graph neural network. The model is motivated by their observation that filtration process in persistent homology aligns with the graph pooling process, thus if the reduced graphs share similar topological information with the original graph, the pool process will be efficient. They characterize the topology of the graphs by using persistent homology and design a corresponding topological loss term to guide the graph pooling. They have incorporated the TIP model into a collection of graph pooling methods. The results are very promising.

**Strengths:**

The key innovation of the paper is the design a topology-based pooling process. Different from previous pooling process when preserving the cluster/community structure is the main focus, the TIP model is to maintain the topological similarities in terms of persistent homology information, during the graph pooling process.

**Weaknesses:**

Even though the authors claim that “The core of PH is the computation of filtration”, they have not provided a convincing argument of their choice of Forman-Ricci curvature as the filtration parameter. The notations in the paper is confusing and need to be improved.

**Questions:**

1)The authors use Forman-Ricci curvature as the filtration parameter and their reason is that it “incorporates edge weights and graph clusters to better capture the topological features of the coarsened graphs”. In fact, Forman-Ricci curvature is one type of discrete Ricci curvature models. The other discrete Ricci curvature model is Ollivier-Ricci curvature (thus it may be better to call it Forman-Ricci curvature instead of Forman curvature).  Essentially, all these discrete Ricci curvature models only characterize the “local” geometric information of the graph or simplicial complex. It is not sure why the authors choose to use it as the filtration parameter.

2)The authors state that “The core of PH is the computation of filtration, which presents a challenging task due to its complexity”. What is the meaning of “computation of filtration”? Is it to design a proper filtration parameter, to construct a series of simplicial complexes from the filtration process, or to calculate the persistence of homological generator from the filtration process?

3)Some notations in the paper are very confusing. For instance, Page 3, $V={(i, x_v)}_{v\in 1:n}$, why the notation {i} is needed?; $X\in R^{n*d}$, I guess $d$ is the dimension of the node vector?

4)Page 4, for the “sequence of nested subgraphs”, the subindex is from 0 to n, this implies that nodes are added into the filtration process one by one, which is clearly not the case. To avoid the confusion, it is better to use a different way of notations.

5)Page 5, in equation (5), the operation of min and max is in terms of what? If it is for the elements of the matrix, the min(A^l) will result in a number (scale).

6)Page 5, in equation (6), note that “D_1” is defined to be “ph()”, but in page 4, “ph()” is defined to be collections of  “D_0”, “D_1”, etc. Further, $A^l$ is defined as the Hadamard product of $A^l$ and $D_1[1]- D_1[0]$. Here $A^l$ is a matrix and  $D_1[1]- D_1[0]$ is a vector. They may have very different dimensions.

---

> ### Author Response · Authors · 2023-11-20
>
> **Questions**
>
> > 1. (Weaknesses) Essentially, all these discrete Ricci curvature models only characterize the “local” geometric information of the graph or simplicial complex. It is not sure why the authors choose to use it as the filtration parameter.
>
> While it is true that curvature characterizes local geometric information, we use it solely as the filtration function. Filtration is the fundamental concept in persistent homology (PH), providing a means to capture the graph's global structures [1]. In this study, the objective is to select a fixed filtration function to compute the Wasserstein distance between persistence diagrams, serving as an evaluation metric for their topological similarities. Thus, we opt for utilizing the Forman-Ricci curvature as the filtration function, given its suitability for our scenario involving graphs with edge weights and cluster structures. Consequently, it is better equipped to capture the topological features of the coarsened graphs. Another advantage of using Forman-Ricci curvature is its computational efficiency. While alternative options like the Ollivier-Ricci curvature are also viable, they generally result in a heavier computational workload.
>
> > 2. What is the meaning of “computation of filtration”?
>
> We apologize for the lack of clarity in our presentation. In this section, we aim to convey the challenge of selecting the appropriate filtration for achieving optimal performance. We have revised this part as "The core of PH is notion of filtration, which presents a challenging task to choose the right filtration."
>
> > 3. Some notations in the paper are very confusing.
>
> We have revised them accordingly.
>
> > 4. Page 4, for the “sequence of nested subgraphs”, the subindex is from 0 to n, this implies that nodes are added into the filtration process one by one, which is clearly not the case. To avoid the confusion, it is better to use a different way of notations.
>
> Here the subindex does not have to be continuous from 0 to n. We have revised the notations to eliminate any potential ambiguity.
>
> > 5. Page 5, in equation (5), the operation of min and max is in terms of what? If it is for the elements of the matrix, the min($A^l$) will result in a number.
>
> It is for the elements of the matrix to get the maximum and minimum values of A, so it will result in a scalar. Equation (5) represents a regular min-max normalization process.
>
> > 6. Page 5, in equation (6), the two matrices have very different dimensions.
>
> Please note that the persistence diagram $\mathcal{D}$ is extended to match cardinality of the number of edges. Therefore, Eq. (6) represents the element-wise product of the edges in $\mathbf{A}'^{(l)}$. We apologize for the lack of clarity in our presentation. The notations of Eq. (6) have been updated to eliminate any potential ambiguity.
>
> [1] Christoph Hofer, Florian Graf, Bastian Rieck, Marc Niethammer, and Roland Kwitt. Graph filtration
> learning. In International Conference on Machine Learning, pp. 4314–4323. PMLR, 2020.

---

> > ### Comment · Reviewer_S9nw · 2023-11-22
> > **The response is good**
> >
> > My comments have been well addressed. I have no further questions!

---

### Official Review · Reviewer_r2G4 · 2023-10-28

**Soundness:** 2 fair
**Presentation:** 3 good
**Contribution:** 2 fair
**Rating:** 3
**Confidence:** 4

**Summary:**

The paper addresses an important problem, graph pooling, which is a key component of GNNs for graph classification tasks. The authors aim to integrate topological structural information from graphs into the pooling process. They introduce the Topology-Invariant Pooling model, demonstrating significant improvements compared to conventional pooling methods.

**Strengths:**

1. The idea keeping the topology information during pooling process is novel, and it can address one of the issues in GNNs in graph classification tasks.
2. The results in benchmark datasets are very good.

**Weaknesses:**

1. While the concept is quite promising, the implementation falls short in terms of its strength. The authors have attempted to utilize persistent homology to maintain consistent topological information during the pooling process; however, the approach they propose doesn't align with their intended outcome. To achieve their objective, the integration of persistent homology should be fundamentally different.

2. The theorem is a bit irrelevant here as the existence of a filtration function in this context does not mean much for practical purposes.

3. For classification results, it would provide more valuable insights into the performance of the proposed model if the authors compared it with State-of-the-Art (SOTA) GNN results rather than basic models. This approach would offer a more meaningful assessment of the model's capabilities.

===============

More specific concerns:

4. Use of PH is a bit problematic here. You are using sublevel filtration with a node function as far as I see. Since you are using single filtration, edge weights have no importance here. Therefore, the concepts of resampling and persistence injection, which involve varying the edge weights, may not align well with the method's underlying principles.

5. When employing sublevel filtration with a filtering function 'f,' it's crucial to recognize that the resulting persistence diagram does not reflect the topology of the graph itself. Instead, it depicts the evolution of topological features within the subgraphs (complexes) defined by the filtering function. As a result, the persistence diagram consists of tuples of values from your filtering function.
In the context of graph coarsening, the primary objective is to reduce the graph's size while preserving its coarse topological structure. The sublevel filtration applied to the original graph is irrelevant to this goal and holds no significance in this specific context.

6. Including dataset statistics would enhance the exposition.

**Questions:**

This is more of a remark than a question. The notion of introducing topological information into the pooling process is intriguing, but it does present a certain contradiction. The original graph may possess numerous topological features, yet the objective of graph pooling is to transition to a smaller, more compact graph. As a result, expecting the coarsened graph to retain similar topological information from the original one is not very meaningful, especially when using PH with sublevel filtration.

One potential solution in pursuit of this goal involves using TDA to guide the graph pooling process. However, this approach often demands a completely different filtration type, such as the Vietoris-Rips (or power) filtration, which can be computationally expensive. Balancing the desire for topological preservation with the computational cost remains a significant challenge in this context.

---

> ### Author Response · Authors · 2023-11-20
>
> **Weaknesses**
>
> > 1 and 4. The integration of persistent homology should be fundamentally different. Since you are using single filtration, edge weights have no importance here. Therefore, the concepts of resampling and persistence injection, which involve varying the edge weights, may not align well with the method's underlying principles.
>
> PH is utilized to incorporate topological information into the edge weights, which are then utilized as inputs to the GNN for obtaining node features. As a result, the utilization of PH influences the final node representations, which, in turn, impact the filtration values in the subsequent pooling layer, ultimately resulting in varied performance on downstream tasks.
>
> > 2. The theorem is a bit irrelevant here as the existence of a filtration function in this context does not mean much for practical purposes.
>
> The existence of such filtration function implies that theoretically TIP has stronger expressive power than conventional dense pooling methods (such as DiffPool, MinCutPool, and DMoNPool) when it comes to distinguishing non-isomorphic graphs. It does not generally translate into better predictive performance, though. Expressive power serves as an indicator of the method's ability in graph classification tasks [1]. By employing a learnable filtration function to enhance flexibility, the theorem validates the potential capability of our method [2][3]. Additionally, we apply our mechanism to a collection of graph pooling methods and observe consistent and substantial performance gain over several popular datasets in graph classification tasks, which is consistent with our theorem.
>
> > 3. For classification results, it would provide more valuable insights into the performance of the proposed model if the authors compared it with State-of-the-Art (SOTA) GNN results rather than basic models.
>
> We provide additional comparisons with Graph Substructure Networks (GSN) [4], which is a GNN model with a topologically-aware message passing scheme based on substructure encoding. The results are presented in Table 2 and shown below. GSN is highly related to our method TIP, as they both take the graph topology into consideration.
>
> |             | NCI1               | PROTEINS           | IMDB-BINARY        | IMDB-MULTI         | OGBG-MOLHIV        |
> |--------------|:------------------:|:------------------:|:------------------:|:-------------------:|:-------------------:|
> | GSN          |   83.50 ± 2.00    |   74.59 ± 5.00    |   76.80 ± 2.00    |   52.60 ± 3.60     |   76.06 ± 1.74     |
>
> > 5. The sublevel filtration applied to the original graph is irrelevant to this goal and holds no significance in this specific context.
>
> The sublevel filtration applied to the original graph is utilized to evaluate the topological similarity between the original graph and the coarsened graph within the feature space induced by PH. This leads to the inclusion of a topological loss term in Equation (8) in our design. Consequently, our method guides the pooling process to retain the essential topology of the original graph.
>
> > 6. Including dataset statistics would enhance the exposition.
>
> The dataset statistics have been added in Appendix B.2.
>
> [1] Keyulu Xu, Weihua Hu, Jure Leskovec, and Stefanie Jegelka. How powerful are graph neural
> networks? arXiv preprint arXiv:1810.00826, 2018.
>
> [2] Christoph Hofer, Florian Graf, Bastian Rieck, Marc Niethammer, and Roland Kwitt. Graph filtration
> learning. In International Conference on Machine Learning, pp. 4314–4323. PMLR, 2020.
>
> [3] Max Horn, Edward De Brouwer, Michael Moor, Yves Moreau, Bastian Rieck, and Karsten Borgwardt.
> Topological graph neural networks. arXiv preprint arXiv:2102.07835, 2021.
>
> [4] Giorgos Bouritsas, Fabrizio Frasca, Stefanos Zafeiriou, and Michael M Bronstein. Improving graph
> neural network expressivity via subgraph isomorphism counting. IEEE Transactions on Pattern
> Analysis and Machine Intelligence, 45(1):657–668, 2022.

---

> > ### Author Response · Authors · 2023-11-20
> >
> > **Questions:**
> >
> > > Expecting the coarsened graph to retain similar topological information from the original one is not very meaningful, especially when using PH with sublevel filtration.
> >
> > There is significant empirical evidence that substructures are often intimately related to downstream tasks in chemistry, biology, and social network analysis [1]. Furthermore, cycles in graphs have been receiving increasing attention [1][2][3]. These studies highlight the significance of cycles in graphs and suggest that enhancing GNNs' structure-awareness can enhance their expressivity and usefulness for downstream tasks. Thus, our paper aims to preserve cycle structures during the pooling process. The effectiveness of graph pooling relies heavily on the coarsened graphs. Accordingly, we assume that preserving cycles in the coarsened graphs is beneficial for downstream tasks in network science and bioinformatics fields. In this paper, we utilize persistent homology to incorporate these crucial topological features into graph pooling. Experimental results validate the effectiveness of our model, confirming the validity of our assumption.
> >
> > [1] Giorgos Bouritsas, Fabrizio Frasca, Stefanos Zafeiriou, and Michael M Bronstein. Improving graph
> > neural network expressivity via subgraph isomorphism counting. IEEE Transactions on Pattern
> > Analysis and Machine Intelligence, 45(1):657–668, 2022.
> >
> > [2] Jiaxuan You, Jonathan M Gomes-Selman, Rex Ying, and Jure Leskovec. Identity-aware graph
> > neural networks. In Proceedings of the AAAI conference on artificial intelligence, volume 35, pp.
> > 10737–10745, 2021.
> >
> > [3] Yinan Huang, Xingang Peng, Jianzhu Ma, and Muhan Zhang. Boosting the cycle counting power
> > of graph neural networks with iˆ2-gnns. In The Eleventh International Conference on Learning
> > Representations, 2022.

---

> > > ### Comment · Reviewer_r2G4 · 2023-11-22
> > >
> > > Thank you for your responses. It seems my main questions haven't been addressed. The concept is innovative, but integrating PH in this context must be meaningful. The rationale provided for utilizing PH doesn't seem to align with its execution. Therefore, I'm inclined to maintain my current score.

---

> > > > ### Author Response · Authors · 2023-11-22
> > > >
> > > > Thanks for your quick comments, and I think maybe we misunderstood your concern before. Here we provide detailed explanations to address your concern.
> > > >
> > > > Our motivation primarily stems from the observed deficiency in current graph pooling methods, specifically their inadequate preservation of topological structures, and we can amend this problem through PH. We are not utilizing PH to directly represent the graph's topology, but rather employing it as a guiding principle for the pooling process. Such guidance makes sense as we observe there is an alignment between PH and graph pooling process (Fig. 1(b)). While PH itself does not inherently represent the graph's topology, it introduces the crucial concept of "persistence" to characterize the significance of edges in a topological context. Through persistence injection, akin to a pooling process, edges with low persistence are filtered out, aligning with the idea of graph coarsening.
> > > >
> > > > Moreover, the influence of persistence on message passing within GNNs resembles an attention-like mechanism. This enhances the GNN's focus on meaningful topological structures, contributing to the increased likelihood of preserving the original graph's topology in the coarsened graph.
> > > >
> > > > Our theoretical and experimental results corroborate the rationale behind our approach. The integration of PH significantly augments the expressive power of graph pooling methods to distinguish isomorphic graphs. Figures 3 and 4 underscore that our method indeed enhances topology preservation compared to traditional pooling methods that fail to retain any topology. Notably, in graph classification tasks across multiple datasets, our method consistently demonstrates substantial improvements.

---

### Official Review · Reviewer_adEb · 2023-10-29

**Soundness:** 2 fair
**Presentation:** 3 good
**Contribution:** 2 fair
**Rating:** 5
**Confidence:** 4

**Summary:**

The paper proposes a new graph coarsening method based on persistent homology, aiming to preserve the topology of the coarsened graph. In particular, it adds the topological loss between the coarsened graph and the original graph into the graph pooling loss function. Experiments on several benchmarks show its effectiveness.

**Strengths:**

Using persistent homology to enhance graph coarsening is new to the best of my knowledge.

The proposed model manages to effectively preserve the topological information, as suggested by the empirical validations.

**Weaknesses:**

1. My biggest concern is that it is unclear why we need to preserve the mentioned topological information, e.g., the 1-th Betti Number (number of cycles) during graph coarsening. For example, in a molecule graph, a benzene ring can be represented by a 6-cycle (a cycle with 6 nodes), and thus can be captured by PH. The cycle can be coarsened to a point, which is reasonable since it will preserve the structural information. However, it will lead to a change in the 1-th Betti Number.

2. Another highly related question is the unclear illustration of Figure 1. What is the chosen filtration of PH in Figure 1(a) and Figure 1(b)? If the filtration is the color of the nodes, then the nodes with the same color should appear or disappear at the same time. In addition, in Figure 1(c), which graph does the diagram correspond to, and which cycle/connected component does each persistence point corresponds to? Furthermore, from my perspective, Figure 1(b) denotes that we can use a graph pooing function to capture the change of the PH information since the correspondence is stable regardless of different datasets. In other words, we do not need additional efforts to preserve the mentioned topological information.

3. In the experiment part, since TU datasets are suffering from high standard deviation, I recommend adding more popular benchmarks such as ZINC and the OGB datasets. In addition, the running time should be reported on different benchmarks.

**Questions:**

See Weakness

---

> ### Author Response · Authors · 2023-11-20
>
> **Weaknesses**
>
> > 1. Why we need to preserve the mentioned topological information, e.g., the 1-th Betti Number (number of cycles) during graph coarsening?
>
> In this paper, we leverage persistent homology (PH) to inject global topological invariance into pooling layers. For a molecule with some ring-like topology (e.g. a necklace), we assume that the coarsened graph still preserve this this characteristic, which has the potential to enhance performance in downstream tasks. The visualization and experimental results demonstrate our assumption.  Our assumption is supported by the visualization and experimental results. It is worth noting that the original pooling layers already possess some local structure-preserving capability, mainly owing to their reliance on clustering methods.
>
> > 2.  What is the chosen filtration of PH in Figure 1(a) and Figure 1(b)? In Figure 1(c), which graph does the diagram correspond to?
>
> Figure 1(a) provides an illustration of the hierarchical view of PH, with the colors of nodes representing the filtration. Nodes with the same color exhibit simultaneous appearances, disappearances, or merging into new nodes. In Figure 1(b), the filtration is selected based on the criteria described in [1]. Figure 1(c) offers a simple illustration of persistence diagrams, which are not associated with a specific graph. Our purpose here is to show that topological features, such as cycles and connected components, can be found in different regions of the persistence diagrams, where points located in the off-diagonal portion contribute to non-zero persistence.
>
> > Figure 1(b) denotes that we can use a graph pooing function to capture the change of the PH information since the correspondence is stable regardless of different datasets. In other words, we do not need additional efforts to preserve the mentioned topological information.
>
> The persistence ratio represents only the proportion of non-zero persistence and implies that graph pooling may coarsen or sparsify graphs in a manner similar to PH. However, while PH utilizes persistence to indicate the significance of corresponding topological features, this aspect cannot be captured by graph pooling alone. Our fundamental insight is to preserve essential topological features during the coarsening process through the integration of PH and graph pooling, which relies on the integration of PH and graph pooling.
>
> > 3. I recommend adding more popular benchmarks. In addition, the running time should be reported on different benchmarks.
>
> We provide additional experiments on OGBG-MOLHIV from the OGB datasets. The comparison results are shown in Table 2 and the response to Reviewer XtaZ, where our method can significantly improve the baseline pooling methods. Additionally, to compare the running time, we perform experiments on a platform with an AMD EPYC 7542 CPU and a single NVIDIA 3090 GPU. We utilize the default settings from the graph classification experiments and record the running time of 50 epoches training on different datasets.  The running time (in seconds) is reported in Appendix E.3 and shown below, which reveals that the incorporation of TIP does not impose a substantial computational burden.
>
> |             | NCI1   | PROTEINS | ENZYMES |
> |--------------|:------:|:--------:|:-------:|
> | DiffPool     | 209.48 |  56.55   |  30.61  |
> | DiffPool-TIP | 365.98 | 113.65   |  61.02  |
> | MinCutPool   | 145.99 |  38.22   |  27.34  |
> | MinCutPool-TIP | 341.93 | 102.31   |  55.54  |
> | DMoNPool     | 124.89 |  35.07   |  19.35  |
> | DMoNPool-TIP | 351.42 | 101.86   |  57.91  |
>
> [1] Christoph Hofer, Florian Graf, Bastian Rieck, Marc Niethammer, and Roland Kwitt. Graph filtration
> learning. In International Conference on Machine Learning, pp. 4314–4323. PMLR, 2020

---

> > ### Comment · Reviewer_adEb · 2023-11-22
> >
> > Thanks for your detailed responses. Regarding the results on OGBG-MOLHIV, they do not appear entirely convincing. It's noteworthy that a GIN framework with a virtual node has achieved a score of 77.07$\pm$1.49 (https://ogb.stanford.edu/docs/leader_graphprop/). The score is comparable to or even better than the result of the persistence-enhanced model. Given that the persistence-enhanced model is 2-3 times slower than the original model, I'm not sure whether the trade-off in efficiency is justified, especially when the GIN framework, which is sufficiently efficient, can produce comparable results.

---

> > > ### Author Response · Authors · 2023-11-23
> > >
> > > The model you linked to employs numerous tricks, such as jumping knowledge, virtual nodes, etc., to achieve its effectiveness, and they use 5 layers of GNN, whereas we only used 3 layers. It's important to emphasize that our model aims to enhance pooling from a topological perspective, rather than adding various dataset-specific tricks to improve performance. As mentioned in the implementation details, we refrain from altering their original model structure while integrating TIP with other graph pooling methods. Furthermore, any additional operations inevitably incur extra computational costs, and the computational overhead introduced by our method is entirely acceptable.

---

### Official Review · Reviewer_XtaZ · 2023-11-01

**Soundness:** 1 poor
**Presentation:** 2 fair
**Contribution:** 1 poor
**Rating:** 3
**Confidence:** 4

**Summary:**

The paper proposes a PH-based pooling layer called TIP (topology-invariant pooling). The proposed approach resamples graph connections (after soft-cluster assignments) and scales edge weights with persistence information (from 1-dim persistence diagrams). In addition, TIP applies a new loss that enforces topology-preservation based on multiple vectorizations of the 1-dim diagrams. Experiments on 3 synthetic and 7 real-world datasets aim to show the efficacy of TIP when combined with well-known pooling methods (DiffPool, MinCutPool, and DMoNPool) and GNNs.

**Strengths:**

- To my knowledge, this is the first paper to leverage PH for graph pooling;
- The proposed method is simple and can be easily integrated (as pooling method) into several GNNs.

**Weaknesses:**

- [Theory] Theorem 1 states that distinguishing graphs based on 1-dimensional persistence diagrams is more expressive than 1-WL. **I believe this is false**. For instance, let $G$ and $G'$ be two graphs comprising one and two isolated nodes, respectively. The 1-dimensional diagrams for these graphs are identical (and empty) for any filtration function. However, 1-WL distinguishes graphs with different numbers of nodes. In addition, since the paper leverages filtration functions on node features, for graphs that share the same single feature vector (or same color), 1-dim diagrams cannot go beyond Betti-1, which is clearly less expressive than 1-WL.

- [Methodology] Since 1-dim persistence diagrams have limited persistence (the death times are identical), it seems the proposal mainly relies on cycle-preserving pooling (do the authors agree?). At a conceptual level, I am not convinced that this design choice is a desideratum for general-purpose graph pooling and/or graph classification tasks. Overall, I found the motivation for adopting only 1-dimensional PD weak;

- [Experiments] Due to the lack of a strong/principled motivation, I would expect to see more empirical evidence to support the proposed method. The paper only considers TU Datasets, and therefore does not exploit recent efforts (e.g., OGB) to strengthen the evaluation of GNNs. Moreover, except for the Enzymes dataset, the empirical gains do not look significant (with differences in mean accuracies usually within one std, overall) compared to pooling-free approaches. Lastly, the ablation study should consider other datasets and pooling layers.

**Questions:**

1. Could you elaborate on how PH coarsens graphs (as in Figure 1)?
2. Do the persistence ratios in Fig. 1(b) include 0-dim persistence diagrams? If so, it seems misleading to use Fig(b) as a motivation and only exploit 1-dimensional PH information for pooling.
3. Does the proposed method apply multiple filtration functions? If so, how are the 1-dimensional persistence diagrams processed (or combined)?
4. The paper says: "PH cannot directly extract meaningful topological features from A". Can't we apply edge-level learned filtration functions, where the nested sequence of subgraphs is obtained according to the filtration values at each edge?
5. I believe it would be helpful to see examples of (1-dim) persistence diagrams before and after applying TIP. For instance, I am curious to see the diagrams for the ring and grid2d datasets.
6. In the introduction, the paper says 'in addition to concatenating vectorized PH diagram as supplementary features, ..." However, the methodology section doesn't mention these additional features. Is the proposed approach also using 0-dim persistence information as additional node features (like in TOGL)?
7. Is TIP isomorphism invariant?
8. In section 4, the paper uses vectorized 1-dim diagrams to measure topological similarity. However, Fig. 5 shows learning curves using the Wasserstein distance. I was confused. The same happens with the filtration functions --- Forman curvature vs. learnable filtration function --- it seems the former is only used for evaluation purposes (table 1). These choices should be clarified.


Minor comments and suggestions:
- Page 3: $V=\\{i, x_v\\}$ $\rightarrow$ $V=\\{v, x_v\\}$;
- Page 3: Highlight that the GNN in Eq. (2) is not the same as the one in Eq. (4);
- Page 3/4: Filtration functions are defined as $f: \mathcal{G} \rightarrow \mathbb{R}$. However, later in the same paragraph, it appears as $f(x_v)$. We can also find $f(e)$. Please be precise here.
- Page 4: What would be the case (3) "all other edges will be paired with the maximum value of the filtration"? Each edge either creates a cycle (case 1) or not (case 2), no?!
- In Eq. (6), computing the Hadarmard product between $A (n \times n)$ and $(\mathcal{D}_1[1] - \mathcal{D}_1[0]) (m \times 1)$ might be problematic. I suggest being more precise.
- In Eq. (7), the variable $t$ should appear on the right-hand side of $h_t = \text{transform}(D_1)$.
- What do you mean by persistent sub-topology?
- Give more details about the Forman curvature used as the filtration function.
- The caption of Fig. (6) says: "a pair of isomorphic graphs that cannot distinguished by 1-WL but can be distinguished by TIP". It is not clear what the paper wants to convey here (including the plot and the caption).

---

> ### Author Response · Authors · 2023-11-20
>
> **Weaknesses**
>
> > [Theory] One dimensional persistence diagram is less expressive than 1-WL.
>
> In the resampling process in Section 4.2, we forgot to include some implementation details. In Eq. (5), we add **self-loops** to the graph for the diagonal part of $\mathbf{A}^{\prime (l)}$ . For the off-diagonal part of $\mathbf{A}^{\prime (l)}$, we only resample the upper triangular matrix to ensure symmetry. As mentioned at the end of Section 3, the self-loop edges are assigned a dummy tuple value $(f(e), f(e))$. Therefore, for graphs $G_1$ and $G_2$ comprising one and two isolated nodes respectively, their corresponding 1-dim persistence diagrams contain one and two data points respectively, making them still distinguishable. While there are cases where 1-dim persistence diagram and 1-WL have the same expressive power, there are also cases where 1-dim persistence diagram can distinguish while 1-WL cannot, e.g. $G_1$ is a graph containing two non-overlapping triangles (2 cycles) and $G_2$ is a hexagon (1 cycle).
>     We would like to thank Reviewer XtaZ for pointing out the deficiencies in our proof. To address this, we have made revisions to Section 4.2 to include additional implementation details and Theorem 1 in Section 4.3.
>
> > [Methodology] Does proposed method mainly relies on cycle-preserving pooling? The motivation for adopting only 1-dimensional PD is weak.
>
> Our primary motivation is to preserve cycle structures during the pooling process. This motivation stems from the significant empirical evidence that substructures, such as cycles, are often intimately related to downstream tasks in fields like chemistry, biology, and social network analysis. Additionally, there has been an increasing focus on cycles in graphs, as evidenced by several notable works [1, 2, 3]. These works emphasize the significance of cycles in graphs and highlight the potential for structure-aware GNNs to enhance expressivity and improve performance in downstream tasks.
>
>    The effectiveness of graph pooling heavily relies on the quality of the coarsened graphs. Therefore, we assume that preserving cycles in the coarsened graphs can be beneficial for network science and bioinformatics tasks. In this paper, we leverage persistent homology (PH) to incorporate these essential topological features into graph pooling. Experimental results confirm the efficacy of our model, validating our assumption. Regarding the use of 1-dim persistence diagrams exclusively, our rationale is that the majority of graphs are connected with only one connected component. Additionally, we performed experiments comparing the use of 1-dim persistence diagrams and found no improvements in our model. In response to this, we have enhanced the description of the motivation for retaining cycle structures in Section 4.2.
>
> > [Experiments] More empirical evidence to support the proposed method and detailed ablation study.
>
> We conduct additional experiments on OGBG-MOLHIV from the OGB datasets. The comparison results, presented in Table 2 and shown below, demonstrate that our method significantly enhances the performance of the baseline pooling methods.  It's worth noting that our objective is to enhance graph pooling methods through the alignment of PH with graph pooling. Furthermore, we include additional results on the ablation study in Appendix E.2, where we add ablation study of DiffPool-TIP, MinCutPool-TIP and DMoNPool-TIP on a wider range of datasets.
>
> |             | OGBG-MOLHIV      |
> |--------------|-----------------|
> | GCN          | 75.04 ± 0.84    |
> | GIN          | 76.03 ± 0.84    |
> | GraphSage    | 76.18 ± 1.27    |
> | TOGL         | 77.21 ± 1.33    |
> | Graclus      | 76.81 ± 0.70    |
> | TopK         | 76.28 ± 0.67    |
> | DiffPool     | 76.60 ± 1.04    |
> | DiffPool-TIP | 77.75 ± 1.18    |
> | MinCutPool   | 75.60 ± 0.54    |
> | MinCutPool-TIP | 77.18 ± 0.83   |
> | DMoNPool     | 76.30 ± 1.34    |
> | DMoNPool-TIP | 76.70 ± 0.62    |
>
> [1] Giorgos Bouritsas, Fabrizio Frasca, Stefanos Zafeiriou, and Michael M Bronstein. Improving graph
> neural network expressivity via subgraph isomorphism counting. IEEE Transactions on Pattern
> Analysis and Machine Intelligence, 45(1):657–668, 2022.
>
> [2] Jiaxuan You, Jonathan M Gomes-Selman, Rex Ying, and Jure Leskovec. Identity-aware graph
> neural networks. In Proceedings of the AAAI conference on artificial intelligence, volume 35, pp.
> 10737–10745, 2021.
>
> [3] Yinan Huang, Xingang Peng, Jianzhu Ma, and Muhan Zhang. Boosting the cycle counting power
> of graph neural networks with iˆ2-gnns. In The Eleventh International Conference on Learning
> Representations, 2022.

---

> > ### Author Response · Authors · 2023-11-20
> >
> > **Questions**
> >
> > > 1. Could you elaborate on how PH coarsens graphs (as in Figure 1)?
> >
> > As outlined in Section 3, PH leads to a sequence of nested subgraphs of the form $G = G_n  \supseteq \ldots G_{k} \ldots \supseteq G_0 = \emptyset$, where $G_k = (V_k, E_k)$ is a subgraph of $G$. These nested subgraphs can be interpreted as a form of coarsening, gradually filtering out specific nodes or edges through PH, as visualized in Figure 1(a). Here, what we want to express is that PH and graph pooling both seek to coarsen/sparsify a given graph in a hierarchical fashion: while PH gradually derives persistent sub-topology by adjusting the filtering parameter, GP obtains a sub-graph by performing a more aggressive cut-off.
> >
> > > 2. Do the persistence ratios in Fig. 1(b) include 0-dim persistence diagrams?
> >
> > No, we exclusively utilize 1-dim persistence diagrams.
> >
> > > 3. Does the proposed method apply multiple filtration functions?
> >
> > No, we only use one filtration function. However, our method can be easily extended to utilize multiple filtration functions by slightly modifying the left part of Eq. (6) as $D_1=\frac{1}{T} \sum_{t} \mathrm{ph}(\mathbf{A}^{\prime (l)}, \mathrm{sigmoid}(\Phi_t(\mathbf{X}^{(l)})))$. This equation computes the average of multiple filtration values and incorporates it into the graph topology. Experimental results demonstrate that employing multiple filtrations does not yield improvements and introduces additional computational burden, hence we choose to use only one filtration function.
> >
> > > 4. Can't we apply edge-level learned filtration functions, where the nested sequence of subgraphs is obtained according to the filtration values at each edge?
> >
> > This is an excellent point and it's indeed possible to apply edge-level filtrations. However, our approach aims for broader applicability and scalability. Applying node-level filtration allows for easy extension to edge-level filtration, although the reverse is not necessarily true. Furthermore, utilizing node features under the GNN framework is inherent. It is important to note that our main contribution lies in aligning PH with pooling in an existing framework, rather than developing a new type of filtration. In our work, we adopt the framework proposed in [1][2] to employ learnable node-level filtrations. We acknowledge that the application of edge-level filtrations holds promise as a research direction, and it is an avenue we intend to explore in future work.
> >
> > > 5. I believe it would be helpful to see examples of (1-dim) persistence diagrams before and after applying TIP.
> >
> > We have added a new section, labeled Appendix E.1, to present the persistence diagrams of the ring and grid2d datasets before and after applying TIP. This addition aims to provide a visual representation of the impact of TIP on the datasets.
> >
> > > 6. The methodology section doesn't mention PH as additional features. Is the proposed approach also using 0-dim persistence information as additional node features?
> >
> > This part is elaborated in Section 4.2, where we refer to as ``Persistence Injection". As previously mentioned, we do not utilize 0-dimensional persistence information as supplementary node features.
> >
> > > 7. Is TIP isomorphism invariant?
> >
> > Yes. We have added a proposition in Section 4.3, and the proof can be found in Appendix C.
> >
> > > 8. In section 4, the paper uses vectorized 1-dim diagrams to measure topological similarity. However, Fig. 5 shows learning curves using the Wasserstein distance.
> >
> > In Fig. 5, our objective is to demonstrate the topological similarity between the coarsened graph and the original graph. This similarity is measured using the Wasserstein distance.  To ensure the validity of the evaluation metric, it is important to have a fixed filtration function rather than a learnable one. In this study, we utilize the Forman curvature on each edge of the graph as the fixed filtration. The Forman curvature incorporates edge weights and graph clusters, enabling a more accurate capture of the topological features of the coarsened graphs. Further clarification on this matter has been revised in Section 5.2.
> >
> >
> >
> > [1] Giorgos Bouritsas, Fabrizio Frasca, Stefanos Zafeiriou, and Michael M Bronstein. Improving graph
> > neural network expressivity via subgraph isomorphism counting. IEEE Transactions on Pattern
> > Analysis and Machine Intelligence, 45(1):657–668, 2022.
> >
> > [2] Jiaxuan You, Jonathan M Gomes-Selman, Rex Ying, and Jure Leskovec. Identity-aware graph
> > neural networks. In Proceedings of the AAAI conference on artificial intelligence, volume 35, pp.
> > 10737–10745, 2021.

---

> > > ### Author Response · Authors · 2023-11-20
> > >
> > > **Minor comments and suggestions:**
> > >
> > > > 1. What would be the case (3) "all other edges will be paired with the maximum value of the filtration"? Each edge either creates a cycle (case 1) or not (case 2)?
> > >
> > > Case 1 indicates the edges that create the circle, case 2 indicates the other edges in this circle (but do not create the circle), and case 3 indicates the edges that are not included in any cycles.
> > >
> > > > 2. What do you mean by persistent sub-topology?
> > >
> > > It means substructures that have meaningful topology, such as cycles.
> > >
> > > > 3. The caption of Fig. (6) says: "a pair of isomorphic graphs that cannot distinguished by 1-WL but can be distinguished by TIP". It is not clear what the paper wants to convey here (including the plot and the caption).
> > >
> > > In this context, we aim to illustrate that there are certain scenarios where TIP surpasses 1-WL, thereby demonstrating its superior expressive power in comparison to conventional dense pooling methods such as DiffPool, MinCutPool, and DMoNPool.
> > >
> > > > Other comments on notations and expressions.
> > >
> > > We thank reviewer XtaZ for the valuable comments on some notations and expressions, which greatly contribute to enhancing the readability of our paper. We have revised all of them according to the suggestions.

---

> > > > ### Comment · Reviewer_XtaZ · 2023-11-22
> > > >
> > > > I have read the responses to all reviewers, and I appreciate the effort put into addressing each concern. Overall, the presentation has improved, and the main issue with Theorem 1 was fixed (there is still a minor issue that I mentioned below). However, my concerns were not sufficiently addressed. Some final comments:
> > > >
> > > > 1. At a conceptual level, the need for persistent homology (PH) is not justified. The paper only exploits 1-dim holes, and their persistence is trivial --- the death time is always $\infty$. In other words, the independent cycles never die. Consequently, everything you can achieve with persistence homology can be obtained with the GNN (or vertex-level filtration function) and homology. PH should be used to obtain detailed topological information (e.g., noisy holes), which is not the case here.
> > > >
> > > > 2. I am not fully convinced of the alignment between PH and pooling. While filtrations consist of a sequence of increasing complexes, pooling produces decreasing ones. I found reporting PH as an inverted filtration in Fig. (1) a stretch.
> > > >
> > > > 3. The theoretical aspect of the paper is weak. For instance, Theorem 1 states that distinguishing graphs (with self-loops) based on 1-dim persistence diagrams is at least as expressive as the 1-WL test. However, we can turn it into a 'strictly more expressive' statement easily. The idea is that, by adding self-loops, we create $n$ cycles with birth times equal to vertex-level filtration values. Assuming we leverage 1WL(GNN) to obtain these vertex-level filtration values, distinguishing graphs based on their diagrams is at least as expressive as 1-WL. Additionally, we know that the number of persistence tuples with death = $\infty$ tells us the number of cycles, and there are non-isomorphic graphs with different numbers of cycles that 1-WL cannot distinguish (e.g., two triangles and one hexagon).
> > > >
> > > > 4. Theorem 1 says: "in terms of distinguishing isomorphic graphs". The statement should say 'non-isomorphic graphs' --- we don't distinguish isomorphic graphs. The same confusion appears in Fig. 5, where non-isomorphic graphs are said to be isomorphic (top subplot).
> > > >
> > > > Based on these and other issues I raised in my first review, I would like to keep my initial score.

---

> > > > > ### Author Response · Authors · 2023-11-23
> > > > >
> > > > > Thank you for your valuable comments. We sincerely appreciate your efforts during the review process. The suggestions you provided are very helpful for us to improve the paper. We will seriously consider your final comments.

---

### Meta-Review · Area_Chair_3yD6 · 2023-12-05

**Metareview:**

This work performs an attempt at an integration between persistent homology and pooling operators in graphs. While the work's topic is definitely interesting, and in principle worthy of inclusion at a venue like ICLR, the majority of the reviewers identified important issues in the work's presentation, theoretical foundations and methodology, which were not satisfactorily resolved even after the rebuttal discussion. As a result I recommend rejection for this work at this time, but encourage the Authors to continue pursuing this direction!

**Justification For Why Not Higher Score:**

In spite of one reviewer being in support of acceptance, I find that many important issues have not been properly addressed by the rebuttal, and the Authors would benefit from another revision round. See detailed comments by the Reviewers for further context.

**Justification For Why Not Lower Score:**

N/A

---

### Decision · Program_Chairs · 2024-01-16

Reject